# INTERACTIVE WEAK SUPERVISION: LEARNING USEFUL HEURISTICS FOR DATA LABELING

**Benedikt Boecking**[1]**, Willie Neiswanger**[2]**, Eric P. Xing**[1]**, & Artur Dubrawski**[1]

[1]Carnegie Mellon University
`{boecking,epxing,awd}@cs.cmu.edu`

[2]Stanford University
`neiswanger@cs.stanford.edu`

## ABSTRACT

Obtaining large annotated datasets is critical for training successful machine learning models and it is often a bottleneck in practice. Weak supervision offers a promising alternative for producing labeled datasets without ground truth annotations by generating probabilistic labels using multiple noisy heuristics. This process can scale to large datasets and has demonstrated state of the art performance in diverse domains such as healthcare and e-commerce. One practical issue with learning from user-generated heuristics is that their creation requires creativity, foresight, and domain expertise from those who hand-craft them, a process which can be tedious and subjective. We develop the first framework for interactive weak supervision in which a method proposes heuristics and learns from user feedback given on each proposed heuristic. Our experiments demonstrate that only a small number of feedback iterations are needed to train models that achieve highly competitive test set performance without access to ground truth training labels. We conduct user studies, which show that users are able to effectively provide feedback on heuristics and that test set results track the performance of simulated oracles.

## 1 INTRODUCTION

The performance of supervised machine learning (ML) hinges on the availability of labeled data in sufficient quantity and quality. However, labeled data for applications of ML can be scarce, and the common process of obtaining labels by having annotators inspect individual samples is often expensive and time consuming. Additionally, this cost is frequently exacerbated by factors such as privacy concerns, required expert knowledge, and shifting problem definitions.

Weak supervision provides a promising alternative, reducing the need for humans to hand label large datasets to train ML models (Riedel et al., 2010; Hoffmann et al., 2011; Ratner et al., 2016; Dehghani et al., 2018). A recent approach called data programming (Ratner et al., 2016) combines multiple weak supervision sources by using an unsupervised label model to estimate the latent true class label, an idea that has close connections to modeling workers in crowd-sourcing (Dawid & Skene, 1979; Karger et al., 2011; Dalvi et al., 2013; Zhang et al., 2014). The approach enables subject matter experts to specify *labeling functions (LFs)*—functions that encode domain knowledge and noisily annotate subsets of data, such as user-specified heuristics or external knowledge bases—instead of needing to inspect and label individual samples. These weak supervision approaches have been used on a wide variety of data types such as MRI sequences and unstructured text, and in various domains such as healthcare and e-commerce (Fries et al., 2019; Halpern et al., 2014; Bach et al., 2019; Ré et al., 2020). Not only does the use of multiple sources of weak supervision provide a scalable framework for creating large labeled datasets, but it can also be viewed as a vehicle to incorporate high level, conceptual feedback into the data labeling process.

In data programming, each LF is an imperfect but reasonably accurate heuristic, such as a pre-trained classifier or keyword lookup. For example, for the popular *20 newsgroups* dataset, an LF to identify the class '*sci.space*' may look for the token '*launch*' in documents and would be right about 70% of the time. While data programming can be very effective when done right, experts may spend a significant amount of time designing the weak supervision sources (Varma & Ré, 2018) and must

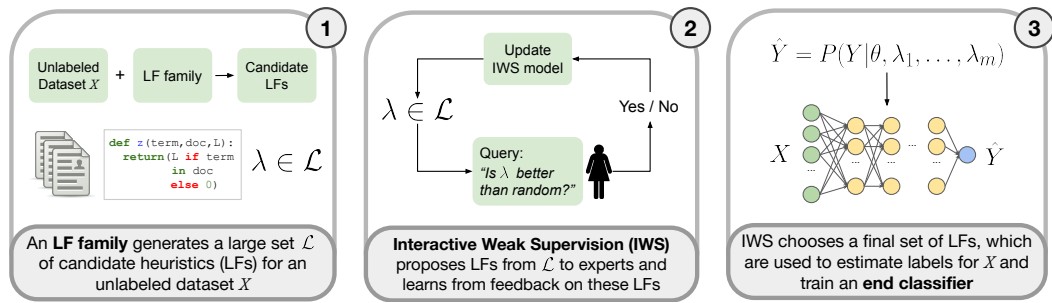

Figure 1: Interactive Weak Supervision (IWS) helps experts discover good labeling functions (LFs).

often inspect samples at random to generate ideas (Cohen-Wang et al., 2019). In our *20 newsgroups* example, we may randomly see a document mentioning '*Salman Rushdie*' and realize that the name of a famous atheist could be a good heuristic to identify posts in '*alt.atheism*'. While such a heuristic seems obvious after the fact, we have to chance upon the right documents to generate these ideas. In practice, coming up with effective LFs becomes difficult after the first few. Substantial foresight (Ramos et al., 2020) is required to create a new function that applies to a non-negligible subset of given data, is novel, and adds predictive value.

We propose a new approach for training supervised ML models with weak supervision through an interactive process, supporting domain experts in fast discovery of good LFs. The method queries users in an active fashion for feedback about candidate LFs, from which a model learns to identify LFs likely to have good accuracy. Upon completion, our approach produces a final set of LFs. We use this set to create an estimate of the latent class label via an unsupervised label model and train a final, weakly supervised end classifier using a noise aware loss function on the estimated labels as in Ratner et al. (2016). The approach relies on the observation that many applications allow for heuristics of varying quality to be generated at scale (similar to Varma & Ré (2018)), and that experts can provide good judgment by identifying some LFs that have reasonable accuracy. The full pipeline of the proposed approach, termed *Interactive Weak Supervision (IWS)*[1], is illustrated in Fig. 1. Our contributions are:

1. We propose, to the best of our knowledge, the first interactive method for weak supervision in which queries to be annotated are not data points but labeling functions. This approach automates the discovery of useful data labeling heuristics.

2. We conduct experiments with real users on three classification tasks, using both text and image datasets. Our results support our modeling assumptions, demonstrate competitive test set performance of the downstream end classifier, and show that users can provide accurate feedback on automatically generated LFs.

3. In our results, IWS shows superior performance compared to standard active learning, i.e. we achieve better test set performance with a smaller number of queries to users. In text experiments with real users, IWS achieves a mean test set AUC after 200 LF annotations that requires at least three times as many active learning iterations annotating data points. In addition, the average user response time for LF queries was shorter than for the active learning queries on data points.

## 2 RELATED WORK

Active strategies for weak supervision sources have largely focused on combinations of data programming with traditional active learning on data points, while our work has similarities to active learning on features (Druck et al., 2009) and active learning of virtual evidence (Lang & Poon, 2021). In Nashaat et al. (2018), a pool of samples is created on which LFs disagree, and active learning strategies are then applied to obtain labels for some of the samples. In Cohen-Wang et al. (2019), samples where LFs abstain or disagree most are selected and presented to users in order to inspire the creation of new LFs. In Hancock et al. (2018), natural language explanations provided during text labeling are used to generate heuristics. The proposed system uses a semantic parser to convert explanations into logical forms, which represent labeling functions.

---

[1]Code is available at `https://github.com/benbo/interactive-weak-supervision`

Prior work has emphasized that LFs defined by experts frequently have a recurring structure in which elements are swapped to change the higher level concept a function corresponds to (Varma & Ré, 2018; Varma et al., 2017; Bach et al., 2019). As an example, in tasks involving text documents, LFs often follow a repetitive structure in which key terms or phrases and syntactical relationships change, e.g. mentions of specific words (Varma & Ré, 2018; Cohen-Wang et al., 2019; Varma et al., 2019). Prior work relies on this observation to create heuristic generators (Varma & Ré, 2018), LF templates (Bach et al., 2019), and domain-specific primitives (Varma et al., 2017). In particular, in a semi-supervised data programming setting, Varma & Ré (2018) propose a system for automatic generation of labeling functions without user interaction, by using a small set of labeled data.

Additional related work has investigated weak supervision for neural networks in information retrieval (Dehghani et al., 2017; Zamani et al., 2018; Zamani & Croft, 2018), the modeling of dependencies among heuristics in data programming (Bach et al., 2017; Varma et al., 2019), the multi-task data programming setting (Ratner et al., 2019), handling of multi-resolution sources (Sala et al., 2019), the use of noisy pairwise labeling functions (Boecking & Dubrawski, 2019), addressing latent subsets in the data (Varma et al., 2016), LFs with noisy continuous scores (Chatterjee et al., 2020), and fast model iteration via the use of pre-trained embeddings (Chen et al., 2020).

## 3 METHODS

We propose an interactive weak supervision (IWS) approach to assist experts in finding good labeling functions (LFs) for training a classifier on datasets without ground truth labels. We will first describe the general problem setting of learning to classify without ground truth samples by modeling multiple weak supervision sources, as well as the concept of LF families. We then dive into the details of the proposed IWS approach. For brevity, we limit the scope of the end classifier to binary classification, but the presented background and ideas do extend to the multi-class settings.

### 3.1 PRELIMINARIES

**Learning with Multiple Weak Supervision Sources**   Assume each data point $x \in \mathcal{X}$ has a latent class label $y^* \in \mathcal{Y} = \{-1, 1\}$. Given $n$ unlabeled, i.i.d. datapoints $X = \{x_i\}_{i=1}^n$, our goal is to train an end classifier $f : \mathcal{X} \to \mathcal{Y}$ such that $f(x) = y^*$. In data programming (Ratner et al., 2016; 2020), a user provides $m$ LFs $\{\lambda_j\}_{j=1}^m$, where $\lambda_j : \mathcal{X} \to \mathcal{Y} \cup \{0\}$. An LF $\lambda_j$ noisily labels the data with $\lambda_j(x) \in \mathcal{Y}$ or abstains with $\lambda_j(x) = 0$. The corresponding LF output matrix is $\Lambda \in \{-1, 0, 1\}^{n \times m}$, where $\Lambda_{i,j} = \lambda_j(x_i)$. In this paper, we assume that each LF $\lambda_j$ has the same accuracy on each class, $\alpha_j = P(\lambda_j(x) = y^* | \lambda_j(x) \neq 0)$, where accuracy is defined on items where $j$ does not abstain. Further, we denote by $l_j = P(\lambda_j(x) \neq 0)$ the LF propensity (sometimes called LF coverage), i.e. the frequency at which LF $j$ does not abstain.

In data programming, an unsupervised label model $p_\theta(Y, \Lambda)$ produces probabilistic estimates of the latent class labels $Y^* = \{y_i^*\}_{i=1}^n$ using the observed LF outputs $\Lambda$ by modeling the LF accuracies, propensities, and possibly their dependencies. A number of label model approaches exist in the crowd-sourcing (Dawid & Skene, 1979; Zhang et al., 2014) and the weak supervision literature (Ratner et al., 2020). In this paper, we use a factor graph as proposed in Ratner et al. (2016; 2020) to obtain probabilistic labels by modeling the LF accuracies via factor $\phi_{i,j}^{Acc}(\Lambda, Y) \triangleq \mathbb{1}\{\Lambda_{ij} = y_i\}$ and labeling propensity by factor $\phi_{i,j}^{Lab}(\Lambda, Y) \triangleq \mathbb{1}\{\Lambda_{ij} \neq 0\}$, and for simplicity assume LFs are independent conditional on $Y$. The label model is defined as

$$p_\theta(Y, \Lambda) \triangleq Z_\theta^{-1} \exp\left(\sum_{i=1}^n \theta^\top \phi_i(\Lambda_i, y_i)\right), \tag{1}$$

where $Z_\theta$ is a normalizing constant and $\phi_i(\Lambda_i, y_i)$ defined to be the concatenation of the factors for all LFs $j = 1, \ldots, m$ for sample $i$. We learn $\theta$ by minimizing the negative log marginal likelihood given the observed $\Lambda$. Finally, following Ratner et al. (2016) an end classifier $f$ is trained using probabilistic labels $p_\theta(Y|\Lambda)$.

**Labeling Function Families**   We define LF families as sets of expert-interpretable LFs described by functions $z_\phi : \mathcal{X} \mapsto \{-1, 0, 1\}$, for parameters $\phi \in \Phi$. An example are shallow decision trees $z_\phi$ parameterized by variables and splitting rules $\phi$ (Varma & Ré, 2018), or a function $z_\phi$ defining a regular expression for two words where $\phi$ parameterizes the word choices from a vocabulary and

the target label. Given such an LF family, we can generate a large set of $p$ candidate heuristics $\mathcal{L} = \{\lambda_j(x) = z_{\phi_j}(x)\}_{j=1}^{p}$, where $\phi_j \in \Phi$, e.g. by sampling from $\Phi$ and pruning low coverage candidates. These families often arise naturally in the form of LFs with repetitive structure that experts write from scratch, where template variables—such as keywords—can be sampled from the unlabeled data to create candidates. For text, we can find n-grams within a document frequency range to generate key term lookups, fill placeholders in regular expressions, or generate shallow decision trees (Ratner et al., 2016; Varma & Ré, 2018; Varma et al., 2019). For time series, we can create a large set of LFs based on motifs (Lonardi & Patel, 2002) or graphs of temporal constraints (Guillame-Bert & Dubrawski, 2017). For images, we can create a library of pre-trained object detectors as in Chen et al. (2019), or in some applications combine primitives of geometric properties of the images (Varma & Ré, 2018).

An LF family has to be chosen with domain expert input. Compared to standard data programming, the burden of creating LFs from scratch is shifted to choosing an appropriate LF family and then judging recommended candidates. We argue that domain experts often have the foresight to choose an LF family such that a sufficiently sized subset of LFs is predictive of the latent class label. Such LF families may not exist for all data types and classification tasks. But when they exist they offer the opportunity to quickly build large, labeled datasets. Once created, it is reasonable to expect that the same LF generation procedure can be reused for similar classification tasks without additional effort (e.g. we use a single LF family procedure for all text datasets in our experiments).

## 3.2 INTERACTIVE WEAK SUPERVISION

Instead of having users provide $m$ good weak supervision sources up front, we want to assist users in discovering them. Successful applications of data programming have established that human experts are able to construct accurate LFs from scratch. Our work leverages the assumption that human experts can also judge these properties when presented with pre-generated LFs of the same form.

Suppose again that we have an unlabeled dataset $X = \{x_i\}_{i=1}^{n}$, and that our goal is to train an **end classifier** $f$ without access to labels $Y^* = \{y_i^*\}_{i=1}^{n}$. Assume also that we defined a large pool of $p$ candidate LFs $\mathcal{L} = \{\lambda_j(x)\}_{j=1}^{p}$ from an LF family (following Sec. 3.1), of varying accuracy and coverage. In IWS, our goal is to identify an optimal subset of LFs $\mathcal{L}^* \subset \mathcal{L}$ to pass to the **label model** in Eq. (1). Below, we will quantify how $\mathcal{L}^*$ depends on certain properties of LFs. While we can observe some of these properties—such as coverage, agreement, and conflicts—an important property that we cannot observe is the accuracy of each LF.

Our goal will thus be to infer quantities related to the latent accuracies $\alpha_j \in [0, 1]$ of LFs $\lambda_j \in \mathcal{L}$, given a small amount expert feedback. To do this, we define an **expert-feedback model**, which can be used to infer LF accuracies given a set of user feedback. To efficiently train this model, our IWS procedure sequentially chooses an LF $\lambda_j \in \mathcal{L}$ and shows a description of $\lambda_j$ to an expert, who provides binary feedback about $\lambda_j$. We follow ideas from active learning for sequential decision making under uncertainty, in which a probabilistic model guides data collection to efficiently infer quantities of interest within $T$ iterations. After a sequence of feedback iterations, we use the expert-feedback model to provide an estimate $\hat{\mathcal{L}} \subset \mathcal{L}$ of the optimal subset $\mathcal{L}^*$. The label model then uses $\hat{\mathcal{L}}$ to produce a probabilistic estimate of $Y^*$, which is used to train the end classifier $f$. The full IWS procedure is illustrated in Fig. 1 and described in detail below.

**Expert-Feedback Model** We first define a generative model of human expert feedback about LFs, given the latent LF accuracies. This model will form the basis for an online procedure that selects a sequence of LFs to show to human experts. We task experts to classify LFs as either useful or not useful $u_j \in \{0, 1\}$, corresponding to their *belief that LF $\lambda_j$ is predictive of $Y^*$ at better than random accuracy for the samples where $\lambda_j$ does not abstain*. Note that prior data programming work (Ratner et al., 2016; 2019; Dunnmon et al., 2020; Saab et al., 2020) assumes and demonstrates that experts are able to use their domain knowledge to make this judgment when creating LFs from scratch. We model the generative process for this feedback and the latent LF accuracies as, for $j = 1, \dots, t$:

$$u_j \sim \text{Bernoulli}(v_j), \quad v_j = h_\omega(\lambda_j), \quad \omega \sim \text{Prior}(\cdot) \tag{2}$$

where $v_j$ can be viewed as the average probability that a human will label a given LF $\lambda_j$ as $u_j = 1$, and $h_\omega(\lambda_j)$ is a parameterized function (such as a neural network), mapping each LF $\lambda_j$ to $v_j$. Finally, to model the connection between accuracy $\alpha_j$ and $v_j$, we assume that $v_j = g(\alpha_j)$, where $g : [0, 1] \to [0, 1]$ is a monotonic increasing function mapping unknown LF accuracy $\alpha_j$ to $v_j$.

After $t$ queries of user feedback on LFs, we have produced a query dataset $Q_t = \{(\lambda_j, u_j)\}_{j=1}^t$. Given $Q_t$, we infer unknown quantities in the above model, which are used to choose the next LF $\lambda_j$ to query, by constructing an acquisition function $\varphi_t : \mathcal{L} \to \mathbb{R}$ and optimizing it over $\lambda \in \mathcal{L}$.

**Acquisition Strategy and Final Set of LFs**   To derive an online procedure for our user queries about LFs, we need to define the properties of the ideal subset of generated LFs $\mathcal{L}^* \subset \mathcal{L}$ which we want to select. Prior data programming work of Ratner et al. (2016; 2019; 2020) with label models as in Eq. (1) does not provide an explicit analysis of ideal metrics of LF sets and their trade-offs to help define this set. We provide the following theorem, which will motivate our definition for $\mathcal{L}^*$.

**Theorem 3.1.** *Assume a binary classification setting, $m$ independent labeling functions with accuracy $\alpha_j \in [0, 1]$ and labeling propensity $l_j \in [0, 1]$. For a label model as in Eq. (1) with given label model parameters $\hat{\theta} \in \mathbb{R}^{2m}$, and for any $i \in \{1, \ldots, n\}$,*

$$P(\hat{y}_i = y_i^*) \geq 1 - \exp\left(-\frac{(\sum_{j=1}^m \hat{\theta}_j^{(1)}(2\alpha_j - 1)l_j)^2}{2||\hat{\theta}^{(1)}||^2}\right)$$

*where $\hat{\theta}^{(1)}$ are the $m$ weights of $\phi^{Acc}$, and $\hat{y}_i \in \{-1, 1\}$ is the label model estimate for $y_i^*$.*

**Proof.**  The proof is given in Appendix C. □

This theorem indicates that one can rank LFs according to $(2\alpha_j - 1)\hat{l}_j$ where $\alpha_j, \hat{l}_j$ are the unknown accuracy and observed coverage of LF $j$, respectively. We provide additional analysis in Appendix C. Our analysis further suggests the importance of obtaining LFs with an accuracy gap above chance. Intuitively, we do not want to add excessive noise by including LFs too close to random. Below, we assume that our final set of LFs is sufficient to accurately learn label model parameters $\hat{\theta}$, and leave analysis of the influence of additional LF properties on learning $\hat{\theta}$ to future work.

To define the ideal final subset of LFs, we distinguish three scenarios: (A) there are no restrictions on the size of the final set and any LF can be included, (B) the final set is limited in size (e.g. due to computational considerations) but any LF can be included, (C) only LFs inspected and validated by experts may be included, e.g. due to security or legal considerations.

For each of these scenarios, at each step $t$ we maximize an acquisition function over the set of candidate LFs, i.e. compute $\lambda_t = \arg\max_{\lambda \in \mathcal{L} \setminus Q_{t-1}} \varphi_t(\lambda)$. We then query a human expert to obtain $(\lambda_t, u_t)$ and update the query dataset $Q_t = Q_{t-1} \cup \{(\lambda_t, u_t)\}$. After a sequence of $T$ queries we return an estimate of $\mathcal{L}^*$, denoted by $\hat{\mathcal{L}}$. The corresponding LF output matrix $\Lambda$ comprised of all $\lambda_j \in \hat{\mathcal{L}}$, is then used to produce an estimate $\widehat{Y}$ of the true class labels via the label model $P_\theta(Y|\Lambda)$. Finally, a noise-aware discriminative end classifier $f$ is trained on $(X, \widehat{Y})$.

*Scenario (A): Unbounded LF Set.* In the absence of restrictions on the final set of LFs, our analysis in Appendix C indicates that the ideal subset of LFs $\mathcal{L}^*$ includes all those with accuracy greater than a gap above chance, i.e. $\alpha_j > r > 0.5$. Thus, we define the optimal subset in this scenario as

$$\mathcal{L}^* = \{\lambda_j \in \mathcal{L} \: : \: \alpha_j > r\}. \tag{3}$$

This is a variation of the task of active level set estimation (LSE), where the goal is to identify all elements in a superlevel set of $\mathcal{L}$ (Zanette et al., 2018; Gotovos, 2013; Bryan et al., 2006). Thus, at each step $t$ we use the straddle acquisition function (Bryan et al., 2006) for LSE, defined for a candidate $\lambda_j \in \mathcal{L} \setminus Q_{t-1}$ to score LFs highest that are unknown and near the boundary threshold $r$:

$$\varphi_t^{\text{LSE}}(\lambda_j) = 1.96 \, \sigma_j(Q_{t-1}) - |\mu_j(Q_{t-1}) - r| \tag{4}$$

where $\sigma_j(Q_{t-1}) = \sqrt{\text{Var}[p(\alpha_j|Q_{t-1})]}$ is the standard deviation and $\mu_j(Q_{t-1}) = \mathbb{E}[p(\alpha_j|Q_{t-1})]$ the mean of the posterior LF accuracy. At the end of Sec. 3.2 we describe how we perform approximate inference of $p(\alpha_j|Q_{t-1})$ via an ensemble model. After a sequence of $T$ queries we return the following estimate of $\mathcal{L}^*$:

$$\hat{\mathcal{L}} = \{\lambda_j \in \mathcal{L} \: : \: \mu_j(Q_T) > r\}. \tag{5}$$

We denote the algorithm for scenario (A) by **IWS-LSE-a**. See Algorithm 1 for pseudocode describing this full IWS-LSE-a procedure. In our experiments, we set $r = 0.7$, though an ablation study shows

that IWS-LSE works well for a range of thresholds $r > 0.5$ (Appendix B.6, Fig. 9). Note that the LSE acquisition function aims to reduce uncertainty around $r$, and therefore tends to explore LFs that have coverage on parts of $Y$ that we are still uncertain about.

*Scenario (B): Bounded LF Set.* If the final set is restricted in size to $m$ LFs, e.g. due to computational considerations when learning the label model in Eq. (1), we need to take the trade-off of LF accuracy and LF coverage into account. Let $\hat{l}_j$ be the observed empirical coverage of LF $\lambda_j$. We want to identify LFs with accuracy above $r$ and rank them according to their accuracy-coverage trade-off, thus our analysis in the appendix suggests the optimal subset is

$$\mathcal{L}^* = \underset{\mathcal{D} \subseteq \mathcal{L}, |\mathcal{D}|=m}{\arg\max} \sum_{\lambda_j \in \mathcal{D}} \left( \mathbb{1}_{\{\alpha_j > r\}} (2 * \alpha_j - 1) * \hat{l}_j \right). \tag{6}$$

Since the LF accuracy-coverage trade-off only comes into effect if $\alpha_j > r$, this yields the same acquisition function $\varphi_t^{\text{LSE}}$ in Eq. (4), and we then select the final set as $\hat{\mathcal{L}} = \{\lambda_j \in \mathcal{D} :$ $\arg\max_{\mathcal{D} \subseteq \mathcal{L}, |\mathcal{D}|=m} \sum_{\lambda_j \in \mathcal{D}} (\mathbb{1}_{\{\mu_j(Q_T) > r\}} (2 * \mu_j(Q_T) - 1) * \hat{l}_j)\}$ which corresponds to a simple thresholding and sorting operation. We denote the algorithm for scenario (B) by **IWS-LSE-ac**.

*Scenario (C): Validated LF Set.* Finally, in some application scenarios, only LFs inspected and validated by experts should be used to estimate $Y^*$, e.g. due to security or legal considerations. An LF $j$ is validated if it is shown to an expert who then responds with $u_j = 1$. This leads to an active search problem (Garnett et al., 2015) where our aim is to identify a maximum number of validated LFs (i.e. $u = 1$) in $\mathcal{L}$ given a budget of $T$ user queries, i.e. to compute

$$\mathcal{L}^*_{\text{AS}} = \underset{\mathcal{D} \subset \mathcal{L}, |\mathcal{D}|=T}{\arg\max} \sum_{\lambda_j \in \mathcal{D}} u_j, \qquad \hat{\mathcal{L}} = \{\lambda_j \in Q_T : u_j = 1\}. \tag{7}$$

As in (Garnett et al., 2015; Jiang et al., 2017), we use a one-step look ahead active search acquisition function defined for a candidate $\lambda_j \in \mathcal{L} \setminus Q_{t-1}$ to be the posterior probability that the usefulness label $u_j$ is positive, i.e. $\varphi_t^{\text{AS}}(\lambda_j) = \mu_j(Q_{t-1})$. We denote the algorithm for scenario (C) by **IWS-AS**.

**Approximate Inference Details** We now describe how we use our expert-feedback model in Eq. (2) to infer $p(\alpha_j|Q_t)$, a quantity used in the acquisition functions and final set estimates. Recall that we defined a generative model of human feedback $u_j$ on query LF $\lambda_j$ with latent variables $v_j$ and $\omega$. We assumed a connection between $v_j$ and the latent LF accuracy $\alpha_j$ via a monotonic increasing function $\alpha_j = g(v_j)$. Similar to existing work on high dimensional uncertainty estimation (Beluch et al., 2018; Chitta et al., 2018), we

---

**Algorithm 1: Interactive Weak Supervision (IWS-LSE-a).**

**Input :** $\mathcal{L}$: set of LFs, $T$: max iterations.

1 $Q_0 \leftarrow \varnothing$
2 **for** $t = 1, 2, \ldots, T$ **do**
3 $\quad$ $\lambda_t \leftarrow \arg\max_{\lambda \in \mathcal{L} \setminus Q_{t-1}} \varphi_t(\lambda)$ $\quad \triangleright$ Eq. (4)
4 $\quad$ $u_t \leftarrow \text{ExpertQuery}(\lambda_t)$
5 $\quad$ $Q_t \leftarrow Q_{t-1} \cup \{(\lambda_t, u_t)\}$
6 **end**
7 $\hat{\mathcal{L}} \leftarrow \{\lambda_j \in \mathcal{L} : \mathbb{E}[p(\alpha_j|Q_T)] > r\} \triangleright$ Eq. (5)

---

use an ensemble $\{\tilde{h}_{\omega^{(i)}}\}_{i=1}^s$ of $s$ neural networks $\tilde{h}_\omega$ with parameters $\omega$ to predict $u_j$ given input $\lambda_j$. To perform this prediction, we need a feature representation $\tau(\lambda_j)$ for LFs that is general and works for any data type and task. To create these features, we use the LF output over our unlabeled dataset $\tau_0(\lambda_j) = (\lambda_j(x_1), \ldots, \lambda_j(x_n))$. We then project $\tau_0(\lambda_j)$ to $d'$ dimensions using PCA for a final feature representation $\tau(\lambda_j)$, which is given as input to each $\tilde{h}_\omega$. Our neural network ensemble can now learn functions $\tilde{h} : \mathbb{R}^{d'} \to [0, 1]$, which map from LF features $\tau(\lambda_j)$ to $v_j = p(u_j = 1|Q_t)$. This yields an ensemble of estimates for $v_j$, and through $g^{-1}$, of $\alpha_j$. These are treated as approximate samples from $p(\alpha_j|Q_t)$, and used to form sample-estimates used in the acquisition functions.

## 4 EXPERIMENTS

Our experiments show that heuristics obtained via a small number of iterations of IWS can be used to train a downstream end classifier $f$ with highly competitive test set performance. We first present results obtained with a simulated IWS oracle instead of human users. Oracle experiments allow us to answer how our method would perform if users had perfect knowledge about LF accuracies. We then show results from a user study on text data in which the query feedback is given by humans. In Appendix B.1 we provide results of a user study on images, using image based LFs. A comprehensive description of the datasets and implementation details can be found in Appendix B.2.

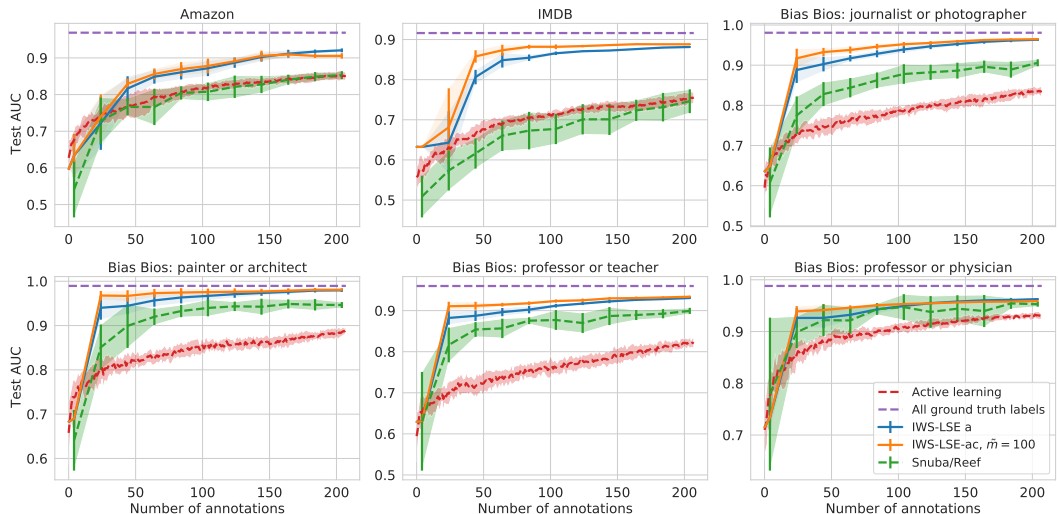

Figure 2: Test set AUC of end classifiers vs. number of iterations. IWS-LSE is compared to active learning, Snuba, and to using all training ground truth labels. Note that one iteration in this plot corresponds to one expert label. A comparison of true user effort needed to answer each type of query (label for one sample vs. label for one LF) will vary by application.

**Datasets**    *Text Classification:*   We create six binary text classification tasks on the basis of three public datasets: Amazon reviews (He & McAuley, 2016), IMDB reviews (Maas et al., 2011), and BiasBios biographies (De-Arteaga et al., 2019). The tasks are chosen such that most English speakers can provide sensible expert feedback on LFs, for ease of reproducibility.
*Cross-modal classification:*   As in Varma & Ré (2018), we take the COCO dataset (Lin et al., 2014) and generate LFs over captions, while classification is performed on the associated images. The two binary tasks are to identify a 'person' in an image, and to identify 'sports' in an image.
*Image classification:*   For image classification tasks with image LFs, we use the COCO dataset and create two binary classification tasks to identify 'sports' in an image and 'vehicle' in an image. For these image-only experiments, we generate nearest-neighbor based LFs directly on the images.

**Approaches**    All approaches train the same downstream end classifier $f$ on the same inputs $X$. We show results for *IWS-LSE-a* (unbounded LF set), *IWS-LSE-ac* (bounded LF set), and *IWS-AS* (validated LF set).   For *IWS-LSE-ac*, we bound the size of the final set of LFs at each iteration $t$ by $m = \sum_{i=1}^{t-1} u_i + \tilde{m}$, i.e. the number of LFs so far annotated as $u = 1$ plus a constant $\tilde{m}$. . We compare the test set performance of IWS to a set of alternatives including (1) annotation of samples via *active learning* (uncertainty sampling) by a noiseless oracle, (2) the *Snuba* system (Varma & Ré, 2018), and (3) using *all ground truth training labels*. In Appendix B.3 we provide additional results comparing IWS to itself using a random acquisition function (*IWS-random*). In our figures, annotations on the x-axis correspond to labeled samples for Snuba and active learning, and to labeled LFs for IWS. We note that this head to head comparison of user effort is application dependent. We provide a timing study to showcase the effort required to carry out IWS versus labeling of samples in our specific user experiments on text in Table 1.

**LF Families**    For text tasks, prior work such as Ratner et al. (2020) and Varma et al. (2019) demonstrates that word and phrase LFs can provide good weak supervision sources. To generate LFs, we define a uni-gram vocabulary over all documents and discard high and low frequency terms. We then exhaustively generate LFs from an LF family $z_\phi$ which outputs a target label if a uni-gram appears in a document, where $\phi$ specifies the uni-gram and target label. We also evaluated combinations of higher-order n-grams, but did not observe a significant change in performance. For COCO images, it is difficult to obtain strong domain primitives to create weak supervision sources, even for data programming from scratch. We hence choose the images and their embeddings themselves to do this job for us, by relying on k-nearest neighbor functions. To generate LFs with high coverage, we first create small, unique clusters of up to $k_1$ mutual nearest neighbors (MkNN)[2].

---

[2]Image A is a $k_1$ nearest neighbor of image B, and image B is also a $k_1$ nearest neighbor of image A.

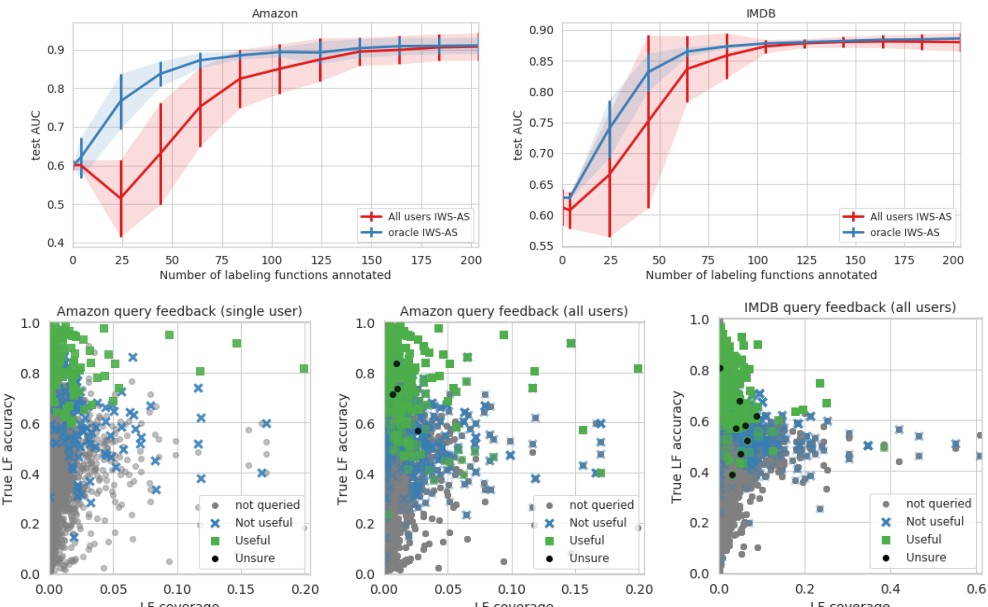

Figure 3: **Human user study, text data.** *Top:* Test AUC of end classifiers trained on soft labels obtained via IWS-AS. Test set performance of humans closely tracks performance using a simulated oracle after ∼100 iterations. *Bottom:* scatter plots of human responses to queries showing the true LF accuracy vs LF coverage by one user (lower left) and all users (lower middle and lower right). An 'unsure' response does not provide a label to an LF query but is counted as an annotation.

For each member of a cluster, we then find the $k_2$ nearest neighbors, and keep ones shared by at least one other cluster member. Finally, each extended cluster defines an LF, which assigns the same label to each member of the extended cluster. The MkNN symmetry produces good initial clusters of varying size, while the second kNN step produces LFs with large and varying coverage. User experiments in Appendix B.1 show that real users can judge the latent LF usefulness quickly by visually inspecting the consistency of the initial cluster and a small selection of the cluster nearest neighbors.

## 4.1 ORACLE EXPERIMENTS

The simulated oracle labels an LF as useful if it has an accuracy of at least 0.7. Measured by test-set AUC of final classifier $f$, IWS-LSE outperforms other approaches significantly on five out of six text datasets, and matches the best performance also attained by Snuba on one dataset (Fig. 2). IWS-AS (Fig. 6, Appendix) performs similarly well on four text datasets, and competitively on the other two. Both IWS approaches outperform active learning by a wide margin on all text datasets. IWS also quickly approaches the performance achieved by an end model trained on the full ground truth training labels. We provide ablation results for IWS-LSE varying the final set size as well as thresholds $r$ in Appendix B.3. For the COCO image tasks, LFs were created using image captions as in (Varma & Ré, 2018) (Fig. 4, first and second plot), as well as on images directly via nearest neighbors (Fig. 4, third and fourth plot). IWS also performs competitively on these image tasks and quickly approaches the performance achieved using all training ground truth.

## 4.2 USER EXPERIMENTS ON TEXT

We conduct experiments of IWS-AS with real users on the Amazon and IMDB review sentiment classification tasks. The results demonstrate that users judge high accuracy functions as useful and make few mistakes. In our experiments, users are shown a description of the heuristic (the key term pattern) and the intended label. Users can also view four snippets of random documents where the LF applied, but are instructed to only consider the examples if necessary. See Appendix B.4 for a screenshot of the query interface and details regarding the user prompts. The top

Table 1: Median (mean) user response time.

| Dataset | Annotate LF | Annotate sample |
|---------|-------------|-----------------|
| Amazon  | 4.2s (8.3s) | 7.9s (10.3s)    |
| IMDB    | 3.2s (6.0s) | 19.s (24.3s)    |

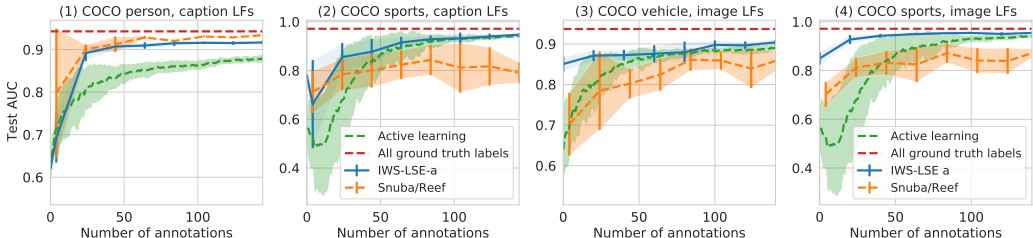

Figure 4: COCO image classification. *Images (1) and (2):* Test AUC of image classifiers trained using probabilistic labels obtained from LFs on captions, compared to training with active learning and the full training ground truth. *Images (3) and (4):* Test AUC of image classifiers trained using nearest neighbor based image LFs compared to training with active learning and the full training ground truth. Due to the low coverage of LFs, we only use IWS-LSE-a in our image experiments.

of Fig. 3 shows that mean test set performance of IWS-AS using LFs obtained from human feedback closely tracks the simulated oracle performance after about 100 iterations. Fig. 3 further shows the queried LFs and corresponding user responses by their true accuracy vs. their non-abstain votes. To match the mean test AUC of IWS-AS obtained after 200 iterations on the Amazon dataset, active learning (uncertainty sampling) requires about 600 iterations. For the IMDB dataset, to achieve the same mean test AUC of IWS-AS obtained after 200 iterations, active learning requires more than 1000 iterations. For both datasets, the average response time to each query was fast. We conducted a manual labeling exercise of samples for the IMDB and Amazon datasets (Table 1) with real users. Assuming the original ratings are true, our users incorrectly classified ∼9% of IMDB reviews while taking significantly longer compared to the response times to LF queries. For the Amazon dataset, users mislabeled ∼2% of samples and were also slower at labeling samples than LFs. The user-study experiments involved nine persons with a computer science background. Neither the true accuracy of each heuristic nor the end model train or test set results were revealed to the users at any stage of the experiment. Appendix B.1 provides results for a similar user study on the COCO sports task with image LFs. These results are consistent with those for text, showing that users are able to distinguish accurate vs. inaccurate image LFs well, and that the full IWS procedure with real users achieves similar performance as the one using a simulated oracle.

## 5 CONCLUSION

We have proposed methods for interactively discovering weak supervision sources. Our results show that a small number of expert interactions can suffice to select good weak supervision sources from a large pool of candidates, leading to competitive end classifiers. The proposed methodology shows promise as a way to significantly speed up the process of weak supervision source discovery by domain experts as an alternative to devising such sources from scratch. On a large number of tasks, we obtain superior predictive performance on downstream test sets compared to the automatic selection of LFs with Snuba (Varma & Ré, 2018) and standard active learning (where users annotate samples instead of LFs), when measured with respect to the number of user annotations. We conduct experiments with real users on two text benchmark datasets and one image dataset and find that humans recognize and approve high accuracy LFs, yielding models that match performance attainable with a simulated oracle. Our text experiments also suggest that tasks exist where users are able to annotate heuristics faster than individual samples.

There are limitations to the approaches we propose in their current form. While our experiments on text and image data show promise, future work is required to investigate appropriate LF families for a wider variety of tasks and data types. Furthermore, we rely on a domain expert's ability to judge the quality of LFs generated from an LF family. While it is true that experts similarly have to judge the quality of LFs they create from scratch, the level of interpretability required of an LF family in IWS may be difficult to achieve for some tasks and data types. Additionally, datasets with a large number of classes may lead to very large sets of candidate LFs. Thus, future work will need to find efficient ways to search this space. Future work should also explore hybrid methods of IWS and active learning, interactive learning of LF dependencies, and acquisition functions which optimize for additional properties of sets of heuristics such as diversity.

ACKNOWLEDGMENTS

We thank Kyle Miller at the Auton Lab for the valuable feedback and discussions. We also want to express our thanks to Maria De Arteaga Gonzalez, Vanessa Kolb, Ian Char, Youngseog Chung, Viraj Mehta, Gus Welter, and Nicholas Gisolfi for their help with this project. This work was partially supported by a Space Technology Research Institutes grant from NASA's Space Technology Research Grants Program, and by Defense Advanced Research Projects Agency's award FA8750-17-2-0130. WN was supported by U.S. Department of Energy Office of Science under Contract No. DE-AC02-76SF00515.

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

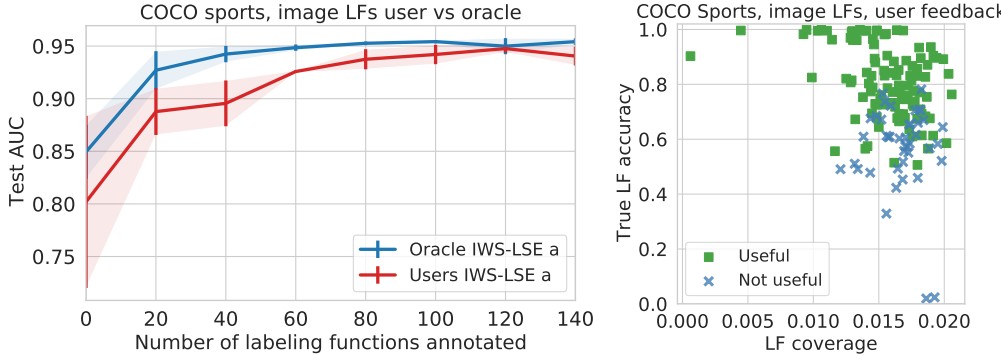

Figure 5: **Human user study, image data (Section B.1).** The user experiments in this plot were done using a labeling function family defined directly on the images. *Left:* Test AUC of end classifiers trained on soft labels obtained via IWS-LSE-a. Test set performance of humans closely tracks performance using a simulated oracle after ∼100 iterations on these datasets. *Right:* scatter plots showing the true LF accuracy vs LF coverage of responses to queries by one user.

## A    PSEUDOCODE FOR DIFFERENT IWS SCENARIOS

---

**Algorithm 2: Interactive Weak Supervision (IWS-LSE-ac).**

**Input :** $\mathcal{L}$: set of LFs, $T$: max iterations.

1  $Q_0 \leftarrow \varnothing$
2  **for** $t = 1, 2, \ldots, T$ **do**
3      $\lambda_t \leftarrow \arg\max_{\lambda \in \mathcal{L} \setminus Q_{t-1}} \varphi_t(\lambda)$  ▷ Eq. (4)
4      $u_t \leftarrow ExpertQuery(\lambda_t)$
5      $Q_t \leftarrow Q_{t-1} \cup \{(\lambda_t, u_t)\}$
6  **end**
7  $\hat{\mathcal{L}} \leftarrow \{\lambda_j \in \mathcal{D} :$
  $\arg\max_{\mathcal{D} \subseteq \mathcal{L}, |\mathcal{D}|=m} \sum_{\lambda_j \in \mathcal{D}} (\mathbb{1}_{\{\mu_j(Q_T) > r\}} (2 *$
  $\mu_j(Q_T) - 1) * \hat{l}_j)\}$

---

**Algorithm 3: Interactive Weak Supervision with Active Search (IWS-AS)**

**Input :** $\mathcal{L}$: set of LFs, $T$: max iterations.

1  $Q_0 \leftarrow \varnothing$
2  **for** $t = 1, 2, \ldots, T$ **do**
3      $\lambda_t \leftarrow \arg\max_{\lambda \in \mathcal{L} \setminus Q_{t-1}} \mu(Q_{t-1})$
4      $u_t \leftarrow ExpertQuery(\lambda_t)$
5      $Q_t \leftarrow Q_{t-1} \cup \{(\lambda_t, u_t)\}$
6  **end**
7  $\hat{\mathcal{L}} \leftarrow \{\lambda_j \in Q_T : u_j = 1\}$

---

Here we provide pseudocode for the IWS-AS and IWS-LSE-ac settings, while the procedure for IWS-LSE-a can be found in the main paper in Algorithm 1. Let us recap why we arrive at different formulations for IWS. In Sec. 3.2, we distinguish three scenarios for arriving at a final set of weak supervision sources which are modeled to obtain an estimate of the latent class variable $Y$. All three scenarios lead to different definitions of an optimal final set of LFs, which in turn means that they require us to formulate appropriate acquisition functions to achieve a good estimate of the optimal set within a budget of $T$ expert interactions. In scenario (A), we place no restrictions on the size of the final set and any LF can be included in it. This means that we have the computational resources to model a potentially very large number of weak supervision sources, and we do not require domain experts to inspect and validate every single LF that is modeled. Importantly, this means that we can and should include LFs that are good according to our predictive model and our definition of the optimal final set of LFs, but have never been shown to an expert. In scenario (B), the final set is limited in size but LFs do not have to be inspected and validated by a user. This scenario may be attractive for rapid cycles during development when a very large number of LFs becomes computationally prohibitive. Finally, in scenario (C), only LFs inspected and validated by experts may be included, e.g. due to security or legal considerations.

## B ADDITIONAL EXPERIMENTS AND RESULTS

### B.1 USER EXPERIMENTS ON IMAGES WITH IMAGE LABELING FUNCTIONS

We carried out a user study on the COCO Sports image classification task described in Section 4, using a family of mutual nearest neighbor image labeling functions, also described in Section 4. In line with our experiments on text data, Figure 5 shows that users were able to judge the accuracy of LFs consistently and well, and that the performance of IWS closely tracks the simulated oracle performance after about 100 iterations.

Again, users were quite quick at responding to LF queries, and judging LFs to be predictive of the latent class variable appeared to be an intuitive task. The average user response time to these image LF queries was 8.8 seconds, while the response time for annotating individual images was around 4.1 seconds on average. To assess an LF, a human user was shown the LFs MkNN image cluster of up to 20 images (the mean size was 7.9 images), and 15 random images contained in the extended cluster, sorted according to their mean distance to the MkNN image cluster. For this nearest neighbor-based family of LFs (as described in Section 4), the parameter $k_1$ was set to 20, and $k_2$ to 1500—though we found that performance was robust to changes in these parameters. While our results show that IWS performs well in this setting, and that classifiers can be trained competitively compared to active learning, it is an interesting challenge to develop better image primitives from which labeling functions can be constructed in data programming, and generated in IWS and Snuba.

### B.2 EXPERIMENT AND IMPLEMENTATION DETAILS

**Datasets** For our text data experiments, we use three publicly available datasets [3] to define six binary text classifaction tasks. We use a subset of the Amazon Review Data (He & McAuley, 2016) for sentiment classification, aggregating all categories with more than $100k$ reviews from which we sample $200k$ reviews and split them into $160k$ training points and $40k$ test points. We use the IMDB Movie Review Sentiment dataset (Maas et al., 2011) which has $25k$ training samples and $25k$ test samples. In addition, we use the Bias in Bios (De-Arteaga et al., 2019) dataset from which we create binary classification tasks to distinguish difficult pairs among frequently occurring occupations. Specifically, we create the following subsets with equally sized train and test sets: journalist or photographer ($n = 32\,258$), professor or teacher ($n = 24\,588$), painter or architect ($n = 12\,236$), professor or physician ($n = 54\,476$).

For the cross-modal tasks of text captions and images as well as the pure image task we use the COCO dataset (Lin et al., 2014). We take the official validation set ($n = 4952$) as the test set. This set of test images is never used at any other point in the pipeline.

**Implementation Details** Our **probabilistic ensemble** in IWS, which is used in all acquisition functions to learn $p(u_j = 1|Q_t)$, is a bagging ensemble of $s = 50$ multilayer perceptrons with two hidden layers of size 10, RELU activations, sigmoid output and logarithmic loss. To create features for the $p$ candidate LFs in $\mathcal{L}$, we use singular value decomposition (SVD) to project from $n$ to $d' = 150$. Thus, at iteration $t$, given a query dataset $Q_{t-1} = \{(\lambda_j, u_j)\}_{j=1}^{t-1}$, the ensemble is trained on pairs $\{(\tau(\lambda_j), u_j)\}_{j=1}^{t-1}$ where $\tau(\lambda_j)$ are the SVD features and $u_j$ the binary expert responses. The output of the ensemble on LFs not in the query dataset is used to compute $\sigma_j(Q_{t-1}) = \sqrt{\mathrm{Var}[g^{-1}(p(u_j = 1|Q_{t-1}))]}$ and $\mu_j(Q_{t-1}) = \mathbb{E}[g^{-1}(p(u_j = 1|Q_{t-1}))]$. While $g$, which maps $\alpha_j$ to $v_j$, could be fine-tuned from data, we set $g$ as the identity function in our experiments, which we find works well empirically. Finally, to allow human experts to express some level of confidence about their decision on $u_j$, we also collect corresponding uncertainty weights $b_j \in \{1, 0.5\}$, and we multiply the contribution to the loss of each $u_j$ by the respective weight $b_j$. Users can also skip queries if they are unsure, indicated in black in Fig. 3. These unsure responses are still counted as an iteration/query in our plots.

Our downstream **end classifier** $f$ is a multilayer perceptron with two hidden layers of size 20 and RELU activations, sigmoid output and logarithmic loss. Each model in the ensemble as well as $f$ are optimized using Adam (Kingma & Ba, 2014). For the text datasets, we fit the end models $f$ to low

---

[3]Amazon: `https://nijianmo.github.io/amazon/index.html`, IMDB: `https://ai.stanford.edu/~amaas/data/sentiment/`, BiasBios: `http://aka.ms/biasbios`

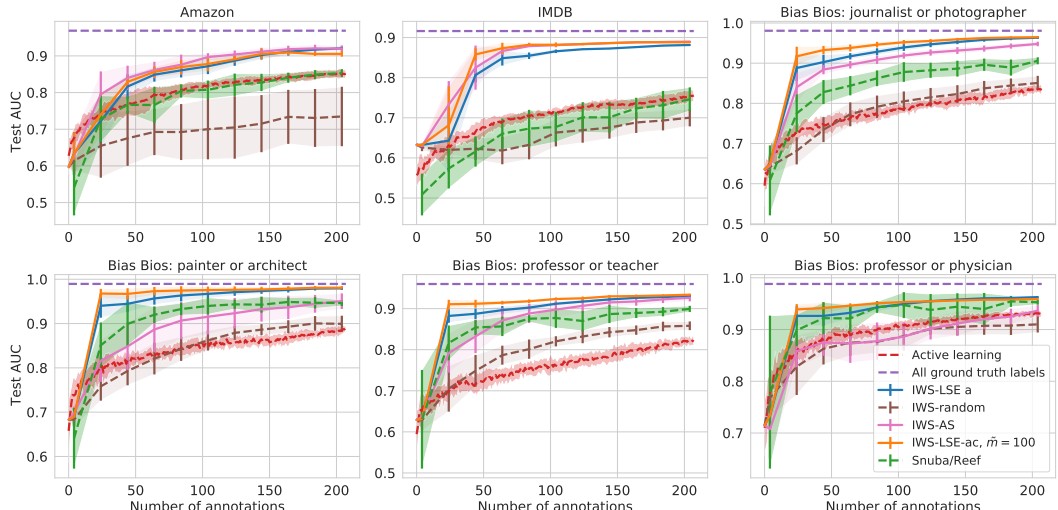

Figure 6: Mean test set AUC vs. number of iterations for end classifiers trained on probabilistic labels. IWS-LSE and IWS-AS are compared to active learning, Snuba, training on all labels, and IWS with a random acquisition function. Note that, while one iteration on this corresponds to one expert label, a comparison of effort needed to answer each type of query (label for sample vs label for LF) will vary by application.

dimensional projections of a large bag-of-words matrix via truncated Singular Value Decomposition (SVD), fixing the embedding size to $d = 300$. We repeat each experiment ten times. We assume that the class balance is known when fitting the label model, as common in related work. When class balance is unknown, (Ratner et al., 2019) discuss an unsupervised approach to estimate it. For the COCO image experiments, we use the second-to-last layer of a ResNet-18 (He et al., 2016) pretrained on ImageNet to obtain image features. These image features are used as the embedding to train the end classifier for all approaches which we compare. The embeddings are also used to create the nearest-neighbor based image LFs.

The first 8 iterations of IWS are initialized with queries of four LFs known to have accuracy between 0.7 and 0.75 drawn at random and four randomly drawn LFs with arbitrary accuracy. Subsequently, IWS chooses the next LFs to query. Active learning is initialized with the same number of known samples.

### B.3 FULL IWS RESULTS AND ALL BASELINES

Fig. 6 provides the full results of all IWS settings to all baselines, including IWS with a random acquisition function (IWS-random). IWS LSE-a corresponds to scenario (A) where there are no restrictions on the size of the final set and any LF can be included]. IWS LSE-ac corresponds to scenario (B) where the final set LFs is limited in size (e.g. due to computational considerations) but any LF can be included. IWS-AS corresponds to Scenario (C), where only LFs in our query dataset $Q_t$ can be used, which are LFs that were inspected and validated by experts, e.g. due to security or legal considerations.

### B.4 USER EXPERIMENTS

#### B.4.1 INTERFACE AND EXPERIMENT PROMPT

Fig. 7 shows an example of the prompt that was shown to users at each iteration of the IWS user experiments. Before the experiment started, users were first instructed on the interface they would see and the task they would be given, i.e. to label a heuristic as good if they would expect it to label samples at better than random accuracy and as bad otherwise. Users were also instructed about the response options, including the option to not answer a query if they were unsure ('I don't know').

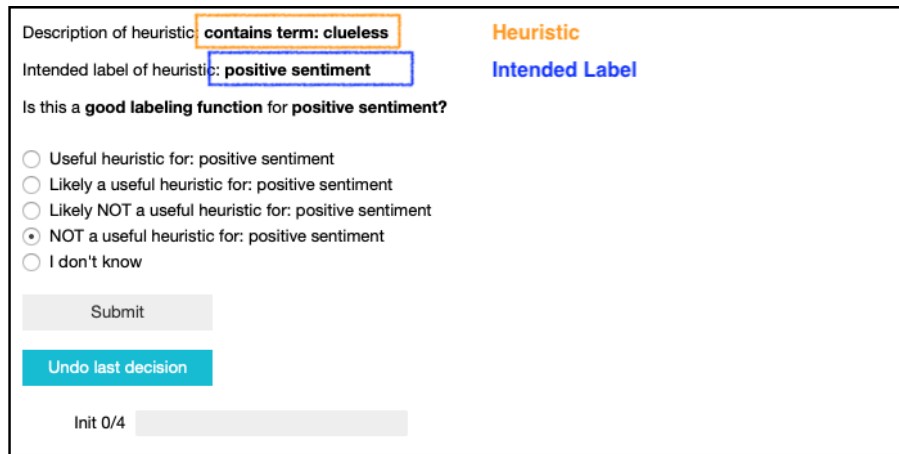

Figure 7: An example of the prompt and answer options that users were shown during the user study. Before starting the experiment, users were provided with a description of the task and the labeling function family.

Users were given a description of the classification task and domain of the documents for which heuristics were being acquired. Users were also provided with a description of the heuristic generated which labeled samples with a target label if a document contained a certain term. Finally, users were given two examples of a better than random heuristic, and two examples of an arbitrary heuristic.

During the experiments, users were also provided with 4 random examples of documents documents where the queried LF applied. Users were instructed to first consider the LF without inspecting these random samples, and to only consider the examples if necessary.

While LFs receive binary labels, users were allowed to express uncertainty about their decision, which was used as a sample weight (1 if certain else 0.5) of LFs during training of the probabilistic model of user feedback.

### B.4.2 ADDITIONAL STATISTICS OF USER EXPERIMENTS

In Fig. 8 we provide more details about our user experiments. The top row displays the test set performance of downstream model $f$ for each individual user. The middle row shows how the number of LFs determined by the user to be useful $u = 1$ increases with the number of iterations. The bottom row displays the maximum positive correlation between a new LF with $u = 1$ at iteration $t$ and all previously accepted LFs with $u = 1$ up to iteration $t$. Note that we take abstains into account by computing correlation between and LF $i$ and $j$ only on entries where at least one of them is nonzero

### B.5 HEURISTICS FOUND

### B.5.1 IWS WITH REAL USERS, SENTIMENT CLASSIFICATION

We here provide some concrete examples of heuristics found during the IWS-AS procedure, that real users annotated as useful during the experiments. For the IMDB dataset:

- Ten terms most frequently annotated as useful by users
  - Class=1: beautiful, wonderful, perfect, enjoyed, amazing, brilliant, fantastic, superb, excellent, masterpiece.
  - Class=0: worst, poor, awful, bad, waste, terrible, horrible, boring, crap, stupid.
- Ten terms annotated as useful with highest underlying accuracy
  - Class=1: flawless, superbly, perfection, wonderfully, captures, refreshing, breathtaking, delightful, beautifully, underrated.
  - Class=0: stinker, dreck, unwatchable, unfunny, waste, atrocious, pointless, redeeming, laughable, lousy.

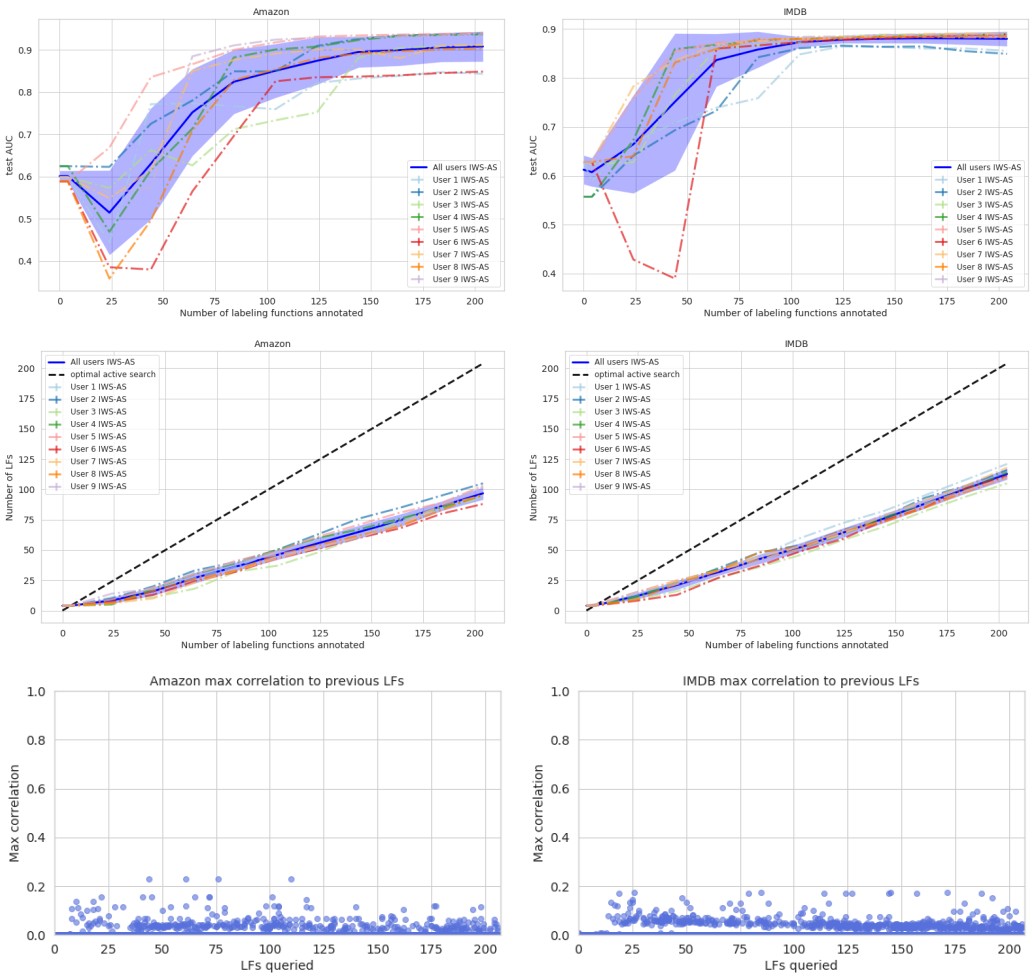

Figure 8: Test AUC vs. IWS iteration shown for individual user experiments with IWS-AS (*top*). Number of LFs labeled as useful vs. IWS iterations (*middle*). Maximum correlation to previously accepted LFs vs. number of iterations (*bottom*).

- Ten terms annotated as useful by users, selected at random
  - Class=1: favorites, joy, superbly, delight, wonderfully, art, intelligent, terrific, light, finest.
  - Class=0: reason, failed, atrocious, decent, unfunny, lame, ridiculous, mistake, worst, dull.

For the Amazon dataset:

- Ten terms most frequently annotated as useful by users
  - Class=1: wonderful, beautiful, amazing, fantastic, favorite, awesome, love, best, perfect, easy.
  - Class=0: worst, terrible, horrible, awful, worse, boring, poor, bad, waste, garbage.
- Ten terms annotated as useful with highest underlying accuracy
  - Class=1: compliments, delightful, pleasantly, stars, captivating, excellent, awesome, beautifully, comfy, perfect.
  - Class=0: poorly, worthless, disappointing, refund, waste, yuck, garbage, unusable, useless, junk.
- Ten terms annotated as useful by users, selected at random
  - Class=1: interesting, beautifully, value, loves, strong, expected, gorgeous, perfectly, durable, great.
  - Class=0: sent, zero, money, mess, crap, refund, wasted, joke, unusable, beware.

### B.5.2  IWS WITH AN ORACLE, OCCUPATION CLASSIFICATION

We here provide examples of heuristics found during the IWS-LSE procedure using a simulated oracle, on the BiasBios biographies datasets. We believe that real users ('internet biography experts') would be able to make very similar distinctions.
For the 'Bias Bios: journalist or photographer' dataset, the ten terms most frequently annotated as useful by the oracle were:

- Class = 1: photography, clients, fashion, studio, photographer, art, commercial, fine, creative, advertising.
- Class = 0: journalism, writing, reporting, news, media, writer, writes, editor, reporter, newspaper.

For the 'Bias Bios: painter or architect' dataset. Ten terms most frequently annotated as useful by oracle:

- Class = 1: commercial, buildings, development, residential, planning, architects, firm, master, design, construction.
- Class = 0: painting, museum, collections, exhibition, gallery, born, artists, shows, series, art.

For the 'Bias Bios: professor or physician' dataset, the ten terms most frequently annotated as useful by the oracle were:

- Class = 1: medical, orthopaedic, residency, family, practice, surgery, memorial, general, physician, saint.
- Class = 0: studies, phd, science, teaching, engineering, received, interests, member, published, professor.

For the 'Bias Bios: professor or teacher' dataset, the ten terms most frequently annotated as useful by the oracle were:

- Class = 1: students, english, teacher, schools, enjoys, years, life, classroom, children, elementary.
- Class = 0: review, research, interests, published, editor, university, journals, associate, studies, phd.

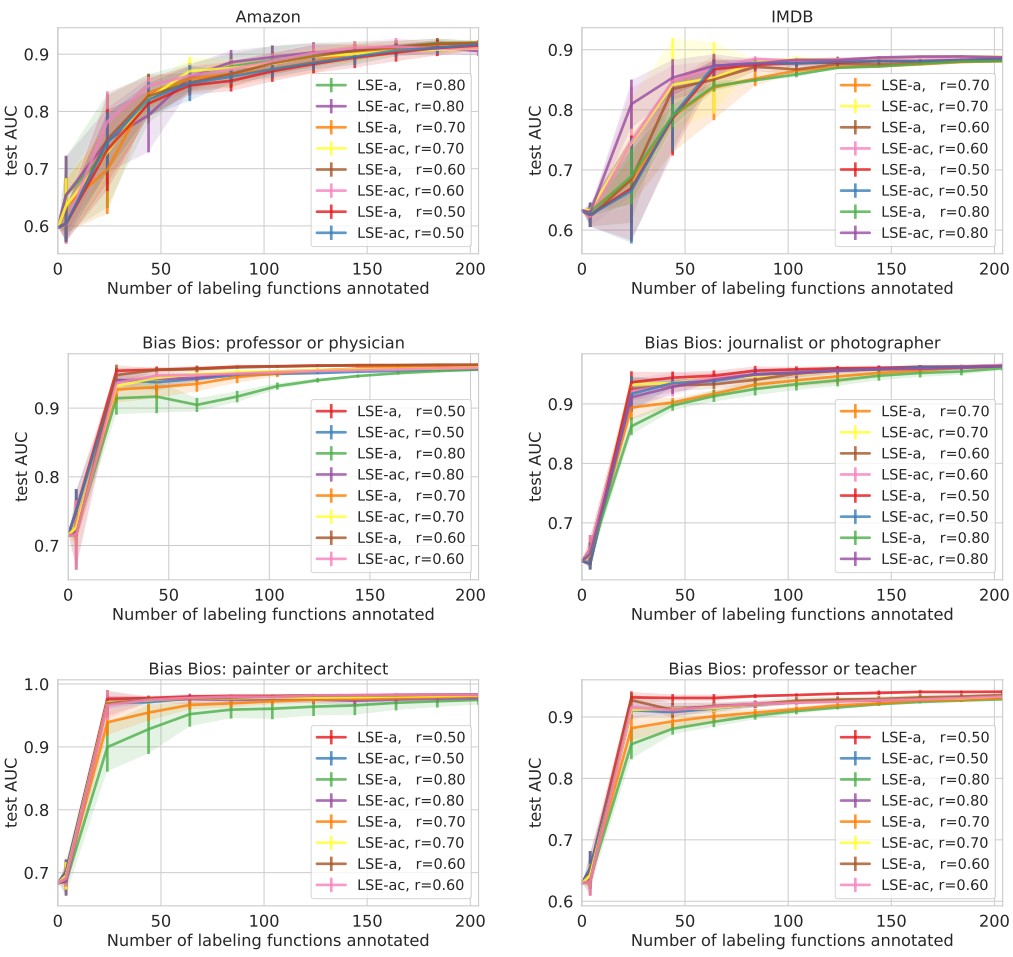

Figure 9: IWS-LSE ablation plots for varying thresholds $r$ which we use to partition our set of LFs. On all datasets test set performance is very similar after around 100 iterations, showing that a wide range of such thresholds leads to good test set performance. For IWS-LSE-ac shown in this plot $\tilde{m}$ was set to 100.

## B.6 ABLATION OF IWS PARAMETER SETTINGS

In this section we provide results of ablation experiments for IWS. The IWS-LSE algorithm requires us to set a threshold $r$ on the (unknown) LF accuracy around which our model aims to partition the set of candidate LFs. Fig. 9 provides results for different $r$ threshold settings for IWS-LSE-a and IWS-LSE-ac, correspondin to Scenario (A) and Scenario (B). The figure shows that the algorithms perform well across a wide range of $r$. While there is no clear, distinct performance difference discernible, the figure suggest that a threshold too close to $1.0$ can cause the algorithm to under-perform. A possible explanation is that as it stifles exploration of LFs within the limited budget of queries to users.

In Scenario (B), which corresponds to the IWS-LSE-ac algorithm, our aim to find a final set of LFs of limited size. Fig. 10 shows that a wide range ($\tilde{m} = 50$ to $200$) of final set sizes produce good results. Recall that in our experiments, we bound the size of the final set of LFs at each iteration $t$ by $m = \sum_{i=1}^{t-1} u_i + \tilde{m}$, i.e. the number of LFs so far annotated as $u = 1$ plus a constant $\tilde{m}$.

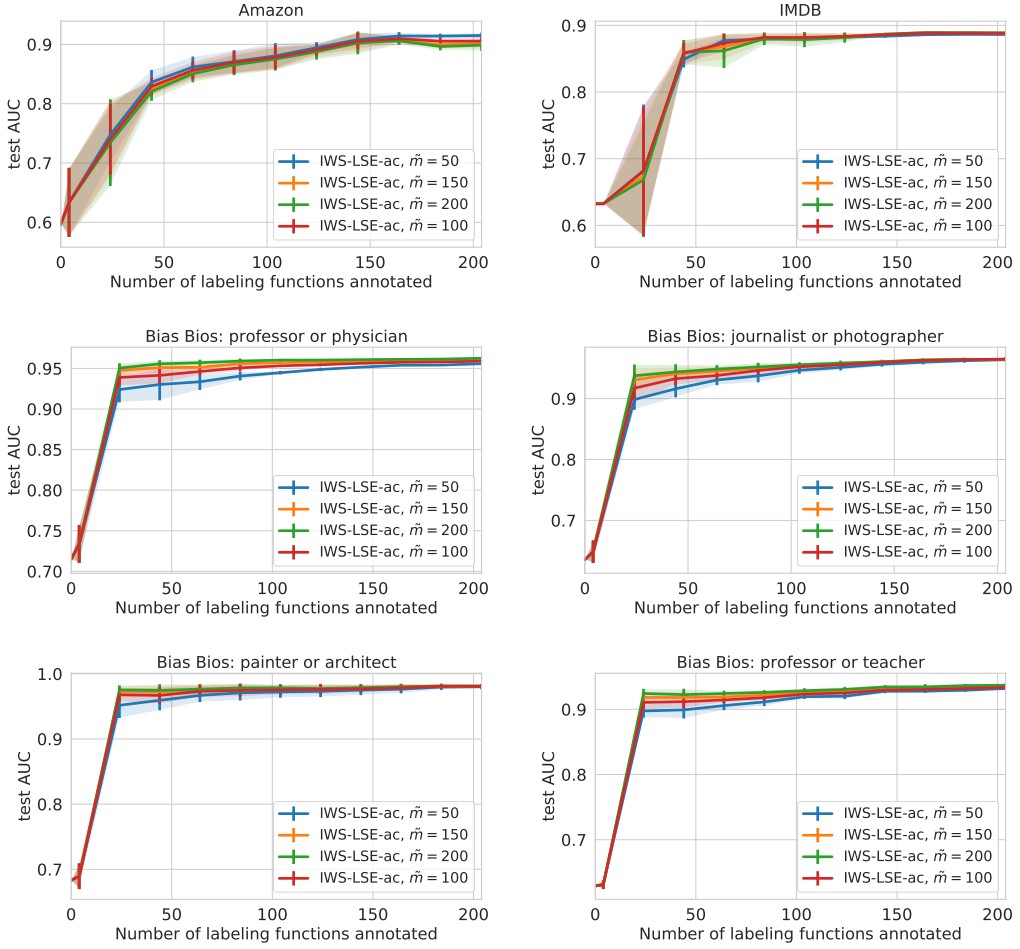

Figure 10: IWS-LSE-ac ablation plots for varying final sizes via parameter $\tilde{m}$. Recall that we bound the size of the final set of LFs at each iteration $t$ by $m = \sum_{i=1}^{t-1} u_i + \tilde{m}$, i.e. the number of LFs so far annotated as $u = 1$ plus a constant $\tilde{m}$. Note that the LSE-ac setting takes LF coverage into account to rank LFs according to $(2\alpha_j - 1) * \hat{l}_j$ where $\alpha_j, \hat{l}_j$ are the estimated LF accuracy and observed LF coverage.

## C    ON LABELING FUNCTION PROPERTIES

In this section we analyze how LF accuracy and LF propensity (i.e. non-abstain behavior) influence the estimate of the true latent class label $Y^*$. We focus on the binary classification case for simplicity. Assume each data point $x \in \mathcal{X}$ has a latent class label $y^* \in \mathcal{Y} = \{-1, 1\}$. Given $n$ unlabeled, i.i.d. data points $X = \{x_i\}_{i=1}^n$, our goal is to train a classifier $f : \mathcal{X} \to \mathcal{Y}$ such that $f(x) = y^*$. As in Ratner et al. (2016) a user provides $m$ LFs $\{\lambda_j\}_{j=1}^m$, where $\lambda_j : \mathcal{X} \to \mathcal{Y} \cup \{0\}$ noisily label the data with $\lambda_j(x) \in \{-1, 1\}$ or abstain with $\lambda_j(x) = 0$. The corresponding LF output matrix is $\Lambda \in \{-1, 0, 1\}^{n \times m}$, where $\Lambda_{i,j} = \lambda_j(x_i)$.

We define a factor graph as proposed in Ratner et al. (2016; 2020) to obtain probabilistic labels by modeling the LF accuracies via factor $\phi_{i,j}^{Acc}(\Lambda, Y) \triangleq \mathbb{1}\{\Lambda_{ij} = y_i\}$ and labeling propensity by factor $\phi_{i,j}^{Lab}(\Lambda, Y) \triangleq \mathbb{1}\{\Lambda_{ij} \neq 0\}$, and for simplicity assume LFs are independent conditional on $Y$. The label model is defined as

$$p_\theta(Y, \Lambda) \triangleq Z_\theta^{-1} \exp\left( \sum_{i=1}^n \theta^\top \phi_i(\Lambda_i, y_i) \right), \tag{8}$$

where $Z_\theta$ is a normalizing constant and $\phi_i(\Lambda_i, y_i)$ defined to be the concatenation of the factors for all LFs $j = 1, \ldots, m$ for sample $i$. Also, let $\theta = (\theta^{(1)}, \theta^{(2)})$ where $\theta^{(1)}, \theta^{(2)} \in \mathbb{R}^m$. Here, $\theta^{(1)}$ are the canonical parameters for the LF accuracies, and $\theta^{(2)}$ the canonical parameters for the LF propensities.

To estimate the label model parameters, we generally obtain the maximum marginal likelihood estimate via the (scaled) log likelihood

$$l(\theta) = 1/n \sum_{i=1}^n \log\left( \sum_{y \in \mathcal{Y}} p(\Lambda_i, y|\theta) \right).$$

Let finite $\hat{\theta} \in \mathbb{R}^{2m}$ be such an estimate. We use $p_{\hat{\theta}}(y|\Lambda_i)$ to obtain probabilistic labels:

$$p_{\hat{\theta}}(y_i = k|\Lambda_i) = \frac{p_{\hat{\theta}}(y_i = k, \Lambda_i)}{p_{\hat{\theta}}(\Lambda_i)} \tag{9}$$

$$= \frac{\exp(\sum_{j=1}^m \hat{\theta}_j^{(1)} \phi^{Acc}(\lambda_j(x_i), k))}{\sum_{\tilde{y} \in \mathcal{Y}} \exp(\sum_{j=1}^m \hat{\theta}_j^{(1)} \phi^{Acc}(\lambda_j(x_i), \tilde{y}))}. \tag{10}$$

Note that the label estimate does not directly depend on $\theta^{(2)}$. Further, note that the denominator is the same over different label possibilities. Finally, note that even in a case where we include correlation factors $\phi_{i,j,k}^{Corr}(\Lambda, Y) = \mathbb{1}\{\Lambda_{ij} = \Lambda_{ik}\}, (j, k) \in C$ in the model above with $C$ as a set of potential dependencies, the probabilistic label will only directly depend on the estimated canonical accuracy parameters $\theta^{(1)}$. In the binary classification case, which we assume here, the expression simplifies further. For $k \in \{-1, 1\}$:

$$p_{\hat{\theta}}(y_i = k|\Lambda_i) = \frac{\exp(\sum_{j=1}^m \hat{\theta}_j^{(1)} \phi^{Acc}(\lambda_j(x_i), k))}{\sum_{\tilde{y} \in \{-1,1\}} \exp(\sum_{j=1}^m \hat{\theta}_j^{(1)} \phi^{Acc}(\lambda_j(x_i), \tilde{y}))} \tag{11}$$

$$= \frac{1}{1 + \exp(\sum_{j=1}^m \hat{\theta}_j^{(1)} (\phi^{Acc}(\lambda_j(x_i), -k) - \phi^{Acc}(\lambda_j(x_i), k)))} \tag{12}$$

$$= \sigma(\sum_{j=1}^m \hat{\theta}_j^{(1)} (\phi^{Acc}(\lambda_j(x_i), k) - \phi^{Acc}(\lambda_j(x_i), -k))), \tag{13}$$

where $\sigma$ denotes the sigmoid function. The probabilistic labels are a softmax in the multi-class classification case and, as shown above, simplify to a sigmoid in the binary case. An absolute label prediction $\hat{y} \in \{-1, 1\}$ is therefore simply a function of

$$\hat{y}_i = \arg\max_{\tilde{y} \in \mathcal{Y}} p_{\hat{\theta}}(y_i = \tilde{y}|\Lambda_i) = \arg\max_{\tilde{y} \in \mathcal{Y}} \sum_{j=1}^m \hat{\theta}_j^{(1)} \phi^{Acc}(\lambda_j(x_i), \tilde{y}).$$

We now introduce some assumptions on the accuracy and error probabilities of labeling functions, similar to the Homogenous Dawid-Skene model (Dawid & Skene, 1979; Li et al., 2013) in crowd sourcing, where label source accuracy is the same across classes and errors are evenly divided with probability mass *independent* of the true class.

Under these assumptions, we denote by $\alpha_j = P(\lambda_j(x) = y^* | \lambda_j(x) \neq 0)$ the accuracy of LF $j$. Further, we denote by $l_j = P(\lambda_j(x) \neq 0)$ the labeling propensity of $j$, i.e. how frequently LF $j$ does not abstain. The observed LF propensity is also referred to as LF coverage in the related literature. We recall theorem 3.1:

**Theorem 3.1.** *Assume a binary classification setting, $m$ independent labeling functions with accuracy $\alpha_j \in [0, 1]$ and labeling propensity $l_j \in [0, 1]$. For a label model as in Eq. (1) with given label model parameters $\hat{\theta} \in \mathbb{R}^{2m}$, and for any $i \in \{1, \ldots, n\}$,*

$$P(\hat{y}_i = y_i^*) \geq 1 - \exp\left(-\frac{(\sum_{j=1}^m \hat{\theta}_j^{(1)}(2\alpha_j - 1)l_j)^2}{2||\hat{\theta}^{(1)}||^2}\right)$$

*where $\hat{\theta}^{(1)}$ are the $m$ weights of $\phi^{Acc}$, and $\hat{y}_i \in \{-1, 1\}$ is the label model estimate for $y_i^*$.*

**Proof.** Assume that we use the label model to obtain a label estimate $\hat{y}_i \in \{-1, 1\}$. As shown in Eq. (10), the prediction rule in that case is

$$\hat{y}_i = \arg\max_{\tilde{y} \in \{-1,1\}} \sum_{j=1}^m \hat{\theta}_j^{(1)} \phi^{Acc}(\lambda_j(x_i), \tilde{y}).$$

Define by $\lambda(x) = (\lambda_1(x), \ldots, \lambda_m(x))$ the vector of the $j = 1, \ldots, m$ LF outputs on $x$. Further, we define for $k \in \{-1, 1\}$:

$$V_{\hat{\theta}}(\lambda(x), k) = \sum_{j=1}^m \hat{\theta}_j^{(1)}(\phi^{Acc}(\lambda_j(x), k) - \phi^{Acc}(\lambda_j(x), -k))$$

$$= \sum_{j=1}^m \hat{\theta}_j^{(1)}\left(\mathbb{1}\{\lambda_j(x) = k\} - \mathbb{1}\{\lambda_j(x) = -k\}\right).$$

For the two label options $k \in \{-1, 1\}$, we have

$$V_{\hat{\theta}}(\lambda(x), 1) = \sum_{j=1}^m \hat{\theta}_j^{(1)}\left(\mathbb{1}\{\lambda_j(x) = 1\} - \mathbb{1}\{\lambda_j(x) = -1\}\right) = \sum_{j=1}^m \hat{\theta}_j^{(1)}\lambda_j(x)$$

and

$$V_{\hat{\theta}}(\lambda(x), -1) = \sum_{j=1}^m \hat{\theta}_j^{(1)}\left(\mathbb{1}\{\lambda_j(x) = -1\} - \mathbb{1}\{\lambda_j(x) = 1\}\right) = -\sum_{j=1}^m \hat{\theta}_j^{(1)}\lambda_j(x).$$

Now, we want to obtain a bound on the probability that the label estimate $\hat{y}_i$ is equal to the true label. We have

$$P(\hat{y}_i = y_i^*) = P(y_i^* = 1)P(\hat{y}_i = 1|y_i^* = 1) + P(y_i^* = -1)P(\hat{y}_i = -1|y_i^* = -1)$$
$$= P(y_i^* = 1)P(\hat{y}_i = 1|y_i^* = 1) + (1 - P(y_i^* = 1))P(\hat{y}_i = -1|y_i^* = -1).$$

Note that

$$P(\hat{y}_i = 1|y_i^* = 1) = P(V_{\hat{\theta}}(\lambda(x_i), 1) > 0|y_i^* = 1) = P(\sum_{j=1}^m \hat{\theta}_j^{(1)}\lambda_j(x) > 0|y_i^* = 1),$$

and that

$$P(\hat{y}_i = -1|y_i^* = -1) = P(V_{\hat{\theta}}(\lambda(x_i), -1) > 0|y_i^* = -1) = P(\sum_{j=1}^{m} \hat{\theta}_j^{(1)} \lambda_j(x_i) < 0|y_i^* = -1).$$

We therefore have

$$P(\hat{y}_i = y_i^*) = P(y_i^* = 1)P(\sum_{j=1}^{m} \hat{\theta}_j^{(1)} \lambda_j(x_i) > 0|y_i^* = 1) + (1 - P(y_i^* = 1))P(\sum_{j=1}^{m} \hat{\theta}_j^{(1)} \lambda_j(x_i) < 0|y_i^* = -1).$$

Now we define $\xi_{ij} = \hat{\theta}_j^{(1)} \lambda_j(x_i)$ and we know that $\xi_j \in [-|\hat{\theta}_j^{(1)}|, |\hat{\theta}_j^{(1)}|]$. Given the Dawid-Skene model assumptions stated previously, we have

$$\mathbb{E}[\sum_{j=1}^{m} \xi_{ij}|y_i^* = 1] = \sum_{j=1}^{m} \mathbb{E}[\xi_{ij}|y_i^* = 1] = \sum_{j=1}^{m} \hat{\theta}_j^{(1)} l_j(2 * \alpha_j - 1),$$

and

$$\mathbb{E}[\sum_{j=1}^{m} \xi_{ij}|y_i^* = -1] = \sum_{j=1}^{m} \mathbb{E}[\xi_{ij}|y_i^* = -1] = -\sum_{j=1}^{m} \hat{\theta}_j^{(1)} l_j(2 * \alpha_j - 1).$$

Now, using Hoeffding's inequality and assuming independent labeling functions, we can bound $P(\hat{y}_i = 1|y_i^* = 1)$ and $P(\hat{y}_i = -1|y_i^* = -1)$ from below:

$$P(\sum_{j=1}^{m} \hat{\theta}_j^{(1)} \lambda_j(x_i) > 0|y_i^* = 1) = P(\sum_{j=1}^{m} \xi_{ij} > 0|y_i^* = 1)$$

$$= P(\sum_{j=1}^{m} \xi_{ij} - \mathbb{E}[\sum_{j=1}^{m} \xi_{ij}|y_i^* = 1] > -\sum_{j=1}^{m} \hat{\theta}_j^{(1)} l_j(2 * \alpha_j - 1) |y_i^* = 1)$$

$$\geq 1 - \exp\left(-\frac{(\sum_{j=1}^{m} \hat{\theta}_j^{(1)}(2\alpha_j - 1)l_j)^2}{2||\hat{\theta}^{(1)}||^2}\right),$$

and

$$P(\sum_{j=1}^{m} \hat{\theta}_j^{(1)} \lambda_j(x_i) < 0|y_i^* = -1) = P(\sum_{j=1}^{m} \xi_{ij} < 0|y_i^* = -1)$$

$$= P(\sum_{j=1}^{m} \xi_{ij} - \mathbb{E}[\sum_{j=1}^{m} \xi_{ij}|y_i^* = -1] < \sum_{j=1}^{m} \hat{\theta}_j^{(1)} l_j(2 * \alpha_j - 1) |y_i^* = -1)$$

$$\geq 1 - \exp\left(-\frac{(\sum_{j=1}^{m} \hat{\theta}_j^{(1)}(2\alpha_j - 1)l_j)^2}{2||\hat{\theta}^{(1)}||^2}\right).$$

Finally we have

$$P(\hat{y}_i = y_i^*) = P(y_i^* = 1)P(\hat{y}_i = 1|y_i^* = 1) + (1 - P(y_i^* = 1))P(\hat{y}_i = -1|y_i^* = -1)$$

$$\geq 1 - \exp\left(-\frac{(\sum_{j=1}^{m} \hat{\theta}_j^{(1)}(2\alpha_j - 1)l_j)^2}{2||\hat{\theta}^{(1)}||^2}\right).$$

$\square$

What do the theorem and the quantities analyzed in this section indicate?

- The trade-off between LF accuracy and LF propensity (also referred to as LF coverage) is captured by $(2\alpha_j - 1)l_j$ which allows us to rank LFs if we know the accuracy $\alpha_j$ or can estimate it and use the observed, empirical coverage as an estimate of $l_j$.

- Not surprising, the relation between $\text{sign}(\theta_j)$ and $\alpha_j$ is important. A better than random LF $j$ should have a positive $\theta_j$. This indicates that a gap to randomness is important if we cannot guarantee that we learn $\theta_j$ well, to reduce the chance of obtaining a negative $\theta_j$ for better than random LF $j$, or vice versa.

- Note how the label estimates are obtained in Eq. (10). Increasing the $\theta_j$ of am LF also effectively means reducing the impact other LFs have on a prediction. In particular when $\theta$ estimates are imperfect, a gap to random accuracy of $\alpha_j$ is important to obtain good label estimates. Intuitively, we do not want to add excessive noise by including LFs close to random unless we can guarantee that their parameter estimate is appropriately low and has the correct sign.

## C.1 WHAT ARE THE OPTIMAL LABEL MODEL PARAMETERS?

Here we discuss the optimal $\theta$ parameters for two cases: first for when we assume that the label model factor graph consists only of accuracy factors and second for when this label model also takes LF propensity into account.

### C.1.1 ASSUMING ACCURACY FACTORS ONLY

In the previous section we assumed that we are given estimated $\theta$ parameters. Naturally, we may next ask ourselves what the optimal theta parameters are. Let us start with a simple case. Let $Y$, $H$ be random variables of the class variable and vector of LFs, respectively. Assume that we only model LF accuracy and define $\mathbb{1}_{\{\lambda(x)=y\}}$ to be the element-wise indicator function. Also, assume that the true distribution can be expressed by the following model:

$$P(Y = y, H = \lambda(x); \theta) = \frac{1}{Z} \exp\left(\theta^\top \mathbb{1}_{\{\lambda(x)=y\}}\right) \tag{14}$$

$$Z = \sum_{y, \lambda(x)} \exp\left(\theta^\top \mathbb{1}_{\{\lambda(x)=y\}}\right). \tag{15}$$

Note that $Z$ can be written as

$$
\begin{aligned}
Z &= \sum_{y, \lambda(x) | \lambda(x)_j = y} \exp\left(\theta^\top \mathbb{1}_{\{\lambda(x)=y\}}\right) + \sum_{y, \lambda(x) | \lambda(x)_j \neq y} \exp\left(\theta^\top \mathbb{1}_{\{\lambda(x)=y\}}\right) \\
&= \sum_{y, \lambda(x) | \lambda(x)_j = y} \exp\left(\theta_j + \theta_{-i}^\top \mathbb{1}_{\{\lambda(x)_{-i}=y\}}\right) + \sum_{y, \lambda(x) | \lambda(x)_j \neq y} \exp\left(\theta_{-i}^\top \mathbb{1}_{\{\lambda(x)_{-i}=y\}}\right) \\
&= (e^{\theta_j} + L) \sum_{y, \lambda(x)_{-i}} \exp\left(\theta_{-i}^\top \mathbb{1}_{\{\lambda(x)_{-i}=y\}}\right)
\end{aligned}
$$

where $L$ is the number of classes. Now, note that

$$P(H_j = Y) = \sum_{y, \lambda(x) | \lambda(x)_j = y} P(Y = y, H = \lambda(x)) \tag{16}$$

$$= \sum_{y, \lambda(x) | \lambda(x)_j = y} \frac{1}{Z} \exp\left(\theta^\top \mathbb{1}_{\{\lambda(x)=y\}}\right) \tag{17}$$

$$= \frac{e^{\theta_j}}{Z} \sum_{y, \lambda(x)_{-i}} \exp\left(\theta_{-i}^\top \mathbb{1}_{\{\lambda(x)_{-i}=y\}}\right) \tag{18}$$

Thus, $P(H_j = Y) = \frac{e^{\theta_j}}{e^{\theta_j}+L}$. Which implies

$$\theta_j = \ln\left(\frac{P(H_j = Y)L}{1 - P(H_j = Y)}\right). \tag{19}$$

Further, note that $P(H_j = Y) = P(H_j = Y, H_j \neq 0) = P(H_j = Y | H_j \neq 0)P(H_j \neq 0)$ is a combination of an LF's accuracy and label propensity.

### C.1.2 ASSUMING ACCURACY FACTORS AND LF PROPENSITY FACTORS

We now analyze a slightly more elaborate label model which also models LF propensity. Again, Let $Y, H$ denote the random variables for class labels and LFs, respectively.

$$P(Y = y, H = \lambda(x); \theta, ) = \frac{1}{Z} \exp\left(\theta^{(1)\top} \mathbb{1}_{\{\lambda(x)=y\}} + \theta^{(2)\top} \mathbb{1}_{\{\lambda(x)\neq 0\}}\right) \tag{20}$$

$$Z = \sum_{y,\lambda(x)} \exp\left(\theta^{(1)\top} \mathbb{1}_{\{\lambda(x)=y\}} + \theta^{(2)\top} \mathbb{1}_{\{\lambda(x)\neq 0\}}\right) \tag{21}$$

$Z$ can be written as

$$Z = \sum_{y,\lambda(x)|\lambda(x)_j=y} \exp\left(\theta^{(1)\top} \mathbb{1}_{\{\lambda(x)=y\}} + \theta^{(2)\top} \mathbb{1}_{\{\lambda(x)\neq 0\}}\right) + \sum_{y,\lambda(x)|\lambda(x)_j\neq y} \exp\left(\theta^{(1)\top} \mathbb{1}_{\{\lambda(x)=y\}} + \theta^{(2)\top} \mathbb{1}_{\{\lambda(x)\neq 0\}}\right)$$

$$= \sum_{y,\lambda(x)|\lambda(x)_j=y} \exp\left(\theta_j^{(1)} + \theta_j^{(2)} + \theta_{-i}^{(1)\top} \mathbb{1}_{\{\lambda(x)_{-i}=y\}} + \theta_{-i}^{(2)\top} \mathbb{1}_{\{\lambda(x)_{-i}\neq 0\}}\right)$$

$$+ \sum_{y,\lambda(x)|\lambda(x)_j\neq y, \lambda(x)_j\neq 0} \exp\left(\theta_j^{(2)} + \theta_{-i}^{(1)\top} \mathbb{1}_{\{\lambda(x)_{-i}=y\}} + \theta_{-i}^{(2)\top} \mathbb{1}_{\{\lambda(x)_{-i}\neq 0\}}\right)$$

$$+ \sum_{y,\lambda(x)|\lambda(x)_j=0} \exp\left(\theta_{-i}^{(1)\top} \mathbb{1}_{\{\lambda(x)_{-i}=y\}} + \theta_{-i}^{(2)\top} \mathbb{1}_{\{\lambda(x)_{-i}\neq 0\}}\right)$$

$$= (e^{\theta_j^{(1)}+\theta_j^{(2)}} + (L-1)e^{\theta_j^{(2)}} + 1) \sum_{y,\lambda(x)_{-i}} \exp\left(\theta_{-i}^{(1)\top} \mathbb{1}_{\{\lambda(x)_{-i}=y\}} + \theta_{-i}^{(2)\top} \mathbb{1}_{\{\lambda(x)_{-i}\neq 0\}}\right)$$

The likelihood of a correct vote is given by

$$P(H_j = Y) = \sum_{y,\lambda(x)|\lambda(x)_j=y} P(Y = y, H = \lambda(x)) \tag{22}$$

$$= \sum_{y,\lambda(x)|\lambda(x)_j=y} \frac{1}{Z} \exp\left(\theta^{(1)\top} \mathbb{1}_{\{\lambda(x)=y\}} + \theta^{(2)\top} \mathbb{1}_{\{\lambda(x)\neq 0\}}\right) \tag{23}$$

$$= \frac{e^{\theta_j^{(1)}+\theta_j^{(2)}}}{Z} \sum_{y,\lambda(x)_{-i}} \exp\left(\theta_{-i}^{(1)\top} \mathbb{1}_{\{\lambda(x)_{-i}=y\}} + \theta_{-i}^{(2)\top} \mathbb{1}_{\{\lambda(x)_{-i}\neq y\}}\right) \tag{24}$$

Further note that

$$P(H_j = 0) = \sum_{y,\lambda(x)|\lambda(x)_j=0} P(Y = y, H = \lambda(x))$$

$$= \sum_{y,\lambda(x)|\lambda(x)_j=0} \frac{1}{Z} \exp\left(\theta^{(1)\top} \mathbb{1}_{\{\lambda(x)=y\}} + \theta^{(2)\top} \mathbb{1}_{\{\lambda(x)\neq 0\}}\right)$$

$$= \frac{1}{Z} \sum_{y,\lambda(x)_{-i}} \exp\left(\theta_{-i}^{(1)\top} \mathbb{1}_{\{\lambda(x)_{-i}=y\}} + \theta_{-i}^{(2)\top} \mathbb{1}_{\{\lambda(x)_{-i}\neq y\}}\right)$$

$$= \frac{1}{e^{\theta_j^{(1)}+\theta_j^{(2)}} + (L-1)e^{\theta_j^{(2)}} + 1}$$

So we can express

$$P(H_j = Y|H_j \neq 0) = \frac{P(H_j = Y, H_j \neq 0)}{P(H_j \neq 0)} = \frac{P(H_j = Y)}{P(H_j \neq 0)} = \frac{P(H_j = Y)}{1 - P(H_j = 0)} \tag{25}$$

$$= \frac{e^{\theta_j^{(1)}+\theta_j^{(2)}}}{e^{\theta_j^{(1)}+\theta_j^{(2)}} + (L-1)e^{\theta_j^{(2)}}} \tag{26}$$

$$P(H_j \neq 0) = 1 - \frac{1}{e^{\theta_j^{(1)}+\theta_j^{(2)}} + (L-1)e^{\theta_j^{(2)}} + 1} \tag{27}$$

Solving for $\theta_j^{(1)}$ and $\theta_j^{(2)}$, we find

$$\theta_j^{(1)*} = \ln \left( \frac{(L-1)\alpha_j}{1 - \alpha_j} \right) \tag{28}$$

$$\theta_j^{(2)*} = \ln \left( \frac{(1 - \alpha_j)l_j}{(L-1)(1 - l_j)} \right), \tag{29}$$

where $\alpha_j = P(H_j = Y | H_j \neq 0)$ and $l_j = P(H_j \neq 0)$. Note that in the binary case $\theta_j^{(1)*}$ is positive only when $\alpha_j > 0.5$.

