# OpenReview forum: "Interactive Weak Supervision: Learning Useful Heuristics for Data Labeling"
_ICLR.cc/2021/Conference — ICLR 2021 Poster_

### Official Review · AnonReviewer4 · 2020-10-26
**Experiments may be over simplistic**

**Rating:** 6
**Confidence:** 3

**Review:**

This paper proposes a new approach for active learning by interactively discovering weak supervision. Instead of asking human to annotate data points, the method collects feedback about candidate label functions, from which a model learns to identify promising label functions. With the final set of label functions, they train a classifier with the estimated labels on unlabeled data. They conduct experiments on text classification datasets with both oracle and human feedback, and show a large improvement compared with traditional active learning.

The paper is well-motivated and the proposed active learning strategy is derived based on theoretical guarantees. I have a few questions about the framework:

-- Does the system start training without any knowledge of the classification task (i.e. no label functions)? If so, how can the model distinguish between the candidates in the beginning?

-- Is the expert-feedback model updated after each human feedback? How is the model updated?

I suggest to add a step-by-step description of the system to help the readers understand the pipeline.

The experiments have been performed on several text classification tasks, where the label function family contains only uni-gram indicators. The end classifier is a small MLP on bag-of-words features. I have concerns about its applicability to real-world NLP tasks. Can the proposed method be extended to more complicated (and useful) label functions such as regular expressions? If we use more advanced text classifiers like BERT, can the proposed strategy still help? Since here the feedback from human is just whether a uni-gram is indicative to the label, which can be easily learned by a neural model.

Missing reference:
Srivastava, Shashank, Igor Labutov, and Tom Mitchell. "Learning to Ask for Conversational Machine Learning." EMNLP 2019.
It also seeks human supervision beyond the instance level.

------
After author response:

The authors presented new experiments on image datasets during the rebuttal, which demonstrate the flexibility of the proposed framework. However, all the experiments are conducted on simple tasks and models. It is still unclear whether and how this method can help more practical problems.

Overall, I think this paper presents a nice exploration towards interactive weak supervision. I hope the authors can release their code and experiment details to encourage future work.

---

> ### Author Response · Authors · 2020-11-17
> **Response to AnonReviewer4**
>
> We would like to thank you very much for your thoughtful questions and feedback.
>
> ### New Experiments with More LF Families
> We have expanded the scope of our experimental results to include a greater range of experiments on additional classification tasks, data types, and labeling function types:
> - LFs for images with captions: We have added two experiments on image captions as performed in the Snuba paper (Varma et al. 2018). For these multi-modal experiments, labeling functions are applied to text captions and an end classifier is trained on images.
> - LFs for images: We have added two experiments on image data, using LFs that directly operate on the images, without any external modality such as text captions.
>
> Please note that these results are preliminary and we are still working on obtaining a full set of baseline comparison for all experiments.
>
> For convenience, most of the changes in our updated paper are denoted by blue text.
>
> ### Q1: IWS initialization
> We apologize that this information was difficult to find. We provided it among the implementation details at the end of Appendix A, before appendix A1. We initialize our experiments with 4 random "good" LFs taken from the generated set $\mathcal{L}$  that have an underlying accuracy between 0.7 and 0.75 and arbitrary coverage. We believe that domain experts should generally know  a similar or greater number of “good” LFs to initialize the algorithm with. We next query the user about 4 LFs chosen at random with arbitrary accuracy and coverage. After these initial responses on 8 LFs in $\mathcal{L}$ are known, the algorithm takes over. Please note that these 8 initialization steps are counted as iterations on the x-axis in our plots.
>
> ### Q2: Expert-feedback model update
> The expert-feedback model describes our modeling assumptions about how user feedback is provided. The expert feedback model contains a parameterized function $h_w(\lambda_j)$ which maps an LF $\lambda_j$ to the average probability that a human will label it as predictive of Y at better than random accuracy (u=1). We approximate this function $h_w(\lambda_j)$ via an ensemble of neural networks, and the parameters of these networks are updated at each iteration $t$ of IWS based on the responses assembled so far in the query dataset $Q_t$ . We provide these details in the *Approximate Inference Details* paragraph at the end of Section 3.
>
> ### Q3: Extension to more complex LFs
> Thank you for this question. For text classification tasks, we could indeed use more complicated labeling functions such as regular expressions. We used unigram LFs in our experiments because they were sufficient to work very well for our experiments. The more complex LFs we constructed did not provide any additional benefit, so we stuck with unigram LFs to simplify the user experiments.  However, we argue that our method can be extended to more complicated LFs if the application requires it. For example, it is straightforward to define a regular expression with placeholders for two words which should occur near each other: \bword1\W+(?:\w+\W+){1,6}?word2\b.
> One would then be able to take the corpus vocabulary, create combinations of words, and prune the generated regex LFs according to some frequency thresholds. This is an interesting direction to explore in future work, especially for more complex text classification tasks. We also want to draw your attention to the new experiments on images that we added to our paper, which show that the method is not constrained to only work well with unigram keyword LFs.
>
> ( Response continued below -> )

---

> > ### Author Response · Authors · 2020-11-17
> > **Response to AnonReviewer4, continued**
> >
> > ### Q4: More advanced text classifiers
> > The existing prior work, in particular by Ratner et al., Bach et al. (see some references below) shows that complex downstream classifiers generalize well from noisy labels obtained from multiple, simple weak supervision sources. In our work, while the feedback from a human expert is whether a uni-gram is indicative of the label (the LF), what the end classifier is trained on is the probabilistic label derived from a large number of such LFs weighted by their estimated accuracy. Our MLP classifier generalizes well from this signal and the related work gives evidence that this should hold for other classes of downstream models as well. A very advanced text classifier/neural model should be able to be trained in this fashion. Compared to our experiments, such a model would likely require access to a larger amount of unlabeled data that the LFs are applied to.
> >
> > Alexander J Ratner, Christopher M De Sa, Sen Wu, Daniel Selsam, and Christopher Re. Data programming: Creating large training sets, quickly. In NIPS, pp. 3567–3575, 2016.
> >
> > Alexander Ratner, Braden Hancock, Jared Dunnmon, Frederic Sala, Shreyash Pandey, and Christopher Re.. Training complex models with multi-task weak supervision. In Proceedings of the AAAI Conference on Artificial Intelligence, volume 33, pp. 4763–4771, 2019.
> >
> > Alexander Ratner, Stephen H Bach, Henry Ehrenberg, Jason Fries, Sen Wu, and Christopher Re. Snorkel: Rapid training data creation with weak supervision. The VLDB Journal, 29(2):709–730, 2020.
> >
> > Stephen H Bach, Daniel Rodriguez, Yintao Liu, Chong Luo, Haidong Shao, Cassandra Xia, Souvik Sen, Alex Ratner, Braden Hancock, Houman Alborzi, et al. Snorkel drybell: A case study in deploying weak supervision at industrial scale. In Proceedings of the 2019 International Conference on Management of Data, pp. 362–375, 2019.
> >
> > ### Q5: Missing reference
> >
> > Thank you for providing this reference, we have incorporated it into our related work.

---

> > > ### Author Response · Authors · 2020-11-24
> > > **Additional Update to Submission (User Study on Image Data)**
> > >
> > > As part of our previous update we had added experiments on image data, including the use of image based labeling functions. Now, our second update includes an additional user-study for the new image-based labeling functions on an image classification task. We show results of this user study in Figure 5 in Appendix B.1 (page 13) of our submission. The results are consistent with the findings of our other user experiments on text data included in the original submission. The results show that users are able to distinguish accurate vs. inaccurate labeling functions well, and that the full IWS procedure with real users achieves similar performance as the ones using a simulated oracle.
> > > In addition to these new user experiments, we were also able to evaluate Snuba on image labeling functions using the same kNN image primitives as IWS, and we updated Figure 4 accordingly.

---

### Official Review · AnonReviewer1 · 2020-10-28
**Official Blind Review #1**

**Rating:** 6
**Confidence:** 4

**Review:**

The paper studies a very interesting problem in weak supervision. One of the main challenges with weak supervision approaches is how to generate the weakly labeled data. The paper suggests an interactive framework for generating and validating labeling functions to generate weakly supervised data.

The user experiments are interesting and  provide very good insights about the method but there are several open questions about the experiments:

- The paper defines an LF family based on the existence of unigrams and exhaustively generate LFs from that family. I understand this would work for sentiment classification (the tow tasks the experiments consider). However, it is not clear to me that this would generalized to other text classification tasks (not to mention language tasks beyond text classification)

- It is interesting to measure and show the effort of validating labeling functions vs. labeling samples. However, I am not sure if that results would hold beyond sentiment analysis too. Looking at the appendix, it seems most LFs are positive words such as "wonderful" and "perfect" or negative words such as "poor" or "bad". These will be significantly easier to judge by a human than other text classification tasks (e.g. the "Salman Rushdie" example in the introduction)

- It would be interesting to compare the proposed approach to a setting where a user is asked to generate LFs based on a description of the tasks and a few examples.

- Regarding the comparison with Snuba, Suba uses a small labeled dataset  to generate heuristics. How was that implemented in the experiments?

----------------------------------
Edited after authors responses: I would like to thank the authors for the detailed response and the changes they have made to the paper. While I still have some concerns and questions about generalization to more complex tasks, models and labeling functions, I think the additional experiments demonstrate the value of the proposed framework and opens the door for future work to explore these questions.

---

> ### Author Response · Authors · 2020-11-17
> **Response to AnonReviewer1**
>
> Thank you for your thoughtful comments and questions which we respond to below. Please note that we have updated our paper to include a greater range of experimental results on additional classification tasks (and on new data types), which we also discuss below.
>
> ### Q1: Generalization to other text classification tasks
> We would first like to emphasize that three of our experiments (Figure 2, bottom row of 3 plots) were not sentiment classification tasks, but were experiments to classify occupations from online biographies. However, we do agree with you that the specific LF family we implemented may not generalize well to all other text classification tasks.  The reason we chose this LF family was because we felt it would be a good initial demonstration of the framework, and since it enabled us to conduct user experiments, using a set of humans without specific domain expertise.
>
> Furthermore, in order to provide more evidence of how our method can generalize to other classification tasks, we have carried out additional experiments on a greater range of data types and labeling function families:
> - LFs for images with captions: We have added two experiments on image captions as performed in the Snuba paper (Varma et al. 2018). This provides an additional non-sentiment classification task.
> - LFs for images: We have added two experiments on image data, using LFs that directly operate on the images, without any external modality such as text captions.
>
> We describe details for the new experiments and LFs in Section 4 of our updated paper. Please note that these results are preliminary and we are still working on obtaining a full set of baseline comparison for all experiments.
>
> For convenience, most of the changes in our updated paper are denoted by blue text.
>
> ### Q2: Effort of validating LFs beyond sentiment classification
> Thank you for this comment, which provoked further clarification. On the topic of user effort, we agree that certain LFs or LF families may take more effort for a human to validate. However, our goal is for a domain expert to be the IWS user, and if strong domain expertise exists, we believe the validation will be low-effort in many cases for these users (e.g. similar to sentiment classification for everyday users). Additionally, in cases where it is difficult for a user to identify a good LF that has been proposed, we contend that it is similarly difficult for a user to manually create good LFs, meaning IWS can still provide benefits to standard data programming by aiding the search for good weak supervision sources.
>
> ### Q3: Additional user experiment
> We agree that this is a good idea. We will aim to complete a relevant user experiment, where users manually create LFs, in time for a camera ready version of this paper.
>
> ### Q4: Snuba comparison
> For our Snuba experiments, we did the following: at each iteration, a human labels one additional unlabeled sample from a dataset, and then Snuba is carried out on this updated dataset. This allows us to plot the performance of Snuba with respect to iterations of human annotation, in order to compare against IWS and active learning.
>
> Similar to how IWS requires an LF family to generate LFs, Snuba requires the definition of domain primitives from which heuristics are generated. In our experiments, these primitives used by Snuba are exactly the same as the ones used by us, e.g. in text experiments we used the same vocabulary to generate unigram based heuristics via the same procedure. Thus, the start up cost to Snuba and IWS is equivalent for all our experiments.

---

> > ### Author Response · Authors · 2020-11-24
> > **Additional Update to Submission (User Study on Image Data)**
> >
> > As part of our previous update we had added experiments on image data, including the use of image based labeling functions. Now, our second update includes an additional user-study for the new image-based labeling functions on an image classification task. We show results of this user study in Figure 5 in Appendix B.1 (page 13) of our submission. The results are consistent with the findings of our other user experiments on text data included in the original submission. The results show that users are able to distinguish accurate vs. inaccurate labeling functions well, and that the full IWS procedure with real users achieves similar performance as the ones using a simulated oracle.
> > In addition to these new user experiments, we were also able to evaluate Snuba on image labeling functions using the same kNN image primitives as IWS, and we updated Figure 4 accordingly.

---

### Official Review · AnonReviewer2 · 2020-10-28
**Official Blind Review #2**

**Rating:** 6
**Confidence:** 4

**Review:**

This paper proposes a new framework for interactively selecting labeling heuristics in a weakly supervised setting. The main idea of the proposed approach is to combine weak supervision and active learning. Compared to the previous work which relies on human manually create labeling functions (the abstraction of the weak supervision), this work defines a family of labeling function and uses an active learning method to interactively identify a set of labeling functions that maximizes the utility based on the usefulness by the users. The experiment results showed that the proposed approach outperforms other baseline methods.

Main comments:
The idea of using an active learning approach to select useful labeling functions from a large pool of labeling functions based on usefulness is interesting and the paper presents compelling solutions to a practical problem. However, the paper lacks clarity to help understand the technical contributions. Here are detailed comments:

1. For labeling function family, the authors defined the labeling function family as sets of expert-interpretable weak supervision sources and shared several examples in different domains. However, there is lack of description of how to free the users to generate such labeling function family or how to adapt the existing labeling families to users' applications, which makes it hard for the readers to understand the exact user interaction in the framework. It's crucial to provide more details here.

2. For the final subset of labeling functions, the authors defined the three scenarios. The distinctions of these different scenarios are not clear. What's the relation between the size of labeling functions selected in the framework and the # of iterations to select the labeling functions? There are some other unclear parts:
  -- For scenario (A), if there are no restrictions about labeling functions, it seems like Eq (4) is not necessary since the framework only needs to find all useful labeling functions.
  -- For scenario (B), it seems that this is the only useful case.
  -- For scenario (C), it seems that this is only a special case.
  -- Can you show the algorithmic procedure of LWS-AS? It's a little confusing about relations between the usefullness of LF and LF validated by experts. What's the difference there?
  -- Overall, Section 3 should be clarified to help reader to understand the model.

3. For the experimental evaluation, the proposed method outperforms the baseline on different tasks. There are several unclear parts:
  -- The comparison with Snuba, it's not fair to compare # of iterations for proposed methods with the # of labeled data since in each iteration the proposed method will apply one more labeling function which likely to label more than 1 sample.
  -- For the IWS-LSE top 100, ac, the authors didn't define what top 100 is. If it's $m$, it's not clear why the performance is improving when the # of iterations is greater than 100.
  -- The approach only evaluated on the text classification problem, it would strengthen the paper if the authors can evaluate in another domain, such as image classification. Snuba has some image applications.

Overall, I like the direction and idea, and I think this paper can be a great paper if the authors can clarify how real users interact with the framework and provide a more thorough experimental analysis.

------ Post Rebuttal------

Thanks to the authors for the new experiments and feedback!

The rebuttal addressed most of my comments and increased my score by one point.

---

> ### Author Response · Authors · 2020-11-17
> **Response to AnonReviewer2**
>
> We would first like to thank you for the useful feedback. We have updated our paper to include both a greater range of experimental results (in particular on image data as you suggested), and we have added clarifications to our submission in response to your review. We respond to your questions and comments in detail below.
>
> ### Q1: Labeling function family
> Thank you for pointing out that further details are necessary. We have clarified the user interaction in Section 3.1. Note that, for convenience, most of the changes in our updated paper are denoted by blue text.
>
> ### Q2: Final subsets of labeling functions
> We want to clarify scenarios (A), (B), and (C), and a few other details of Section 3.
>
> Here is a summary of IWS: We start with a large set of LFs $\mathcal{L}$. Each LF has an observed coverage and unknown accuracy. Based on the accuracy and coverage of the LFs, we derive an optimal subset $\mathcal{L}^* \subset \mathcal{L}$ for data programming. At each query, humans give a binary label $u$ to an LF, where $u=1$ means the LF is predictive of Y. Given a set of binary LF labels, we infer the accuracy of all LFs, and thus estimate a good subset for use in data programming. The goal of scenarios (A), (B), and (C) is:
> - (A) Choose LFs to label in order to best estimate the optimal subset.
> - (B) Choose LFs to label in order to best estimate the optimal subset constrained to a given size (for computational reasons).
> - (C) Choose LFs to label in order to identify the greatest number of LFs with u=1 response (here, the final set of LFs may only contain LFs that have been shown to a human, for trust/explainability/legal reasons).
>
> Based on this summary, we now clarify additional questions from the review.
>
> ### Q2a: Relation between the size of labeling functions selected and # of iterations
> - In scenario (C), at iteration $t$, the size of the selected LF set is the number of LFs queried so far with $u=1$, so the size must be less than $t$.
> - In scenario (B), the size of the selected LF set is equal to the chosen constraint size (see our note below about our implementation details).
> - In scenario (A), there is no strict dependence between the size of the selected LF set and the number of iterations: at each iteration, we return an estimate of the optimal LF set, which does not have a fixed size.
>
> ### Q2b: Eq. (4)  in scenario (A)
> Eq. (4) defines an acquisition function. By maximizing Eq. (4) at each iteration $t$, we choose the next LF to show to a human (line 3 in Algorithm 1). This is necessary for scenario (A) as these human responses are used to train a model which predicts LF accuracy, which in turn is needed to estimate the optimal set.
>
> ### Q2c: Relation between “usefulness of LF” and “LF validated by experts”
> For each LF $\lambda_j$, there is a binary usefulness random variable $u_j$. We observe $u_j$ by asking a human if LF $\lambda_j$ is predictive of Y at better than random accuracy. And we say (e.g. in scenario (C)) that an LF is “validated” if it has been shown to the human and labeled as $u_j = 1$. We have now clarified this in our submission.
>
> ### Q2d: Showing algorithmic procedure for IWS-AS:
> Thank you for pointing out that further clarification is needed. As requested, we added pseudocode for the IWS-AS and IWS-LSE-ac procedure to our submission in Appendix A.
>
> ### Q3a: Comparison with Snuba
> The goal of our experiments is to show performance as it relates to user effort. Each point on the x-axis of our plots corresponds to one additional human annotation. For Snuba, this annotation is a label for a sample, and for IWS it is a label for an LF. We acknowledge in the paper that a direct comparison between IWS and Snuba is difficult: the user effort required to annotate samples compared to the effort required to annotate LFs will heavily depend on the application. However, our text classification experiments show that tasks exist where the annotation of text LFs can be much faster than the annotation of text samples (documents).
>
> Regarding additional user effort required to use either method: similar to how IWS requires an LF family to generate LFs, Snuba requires the definition of domain primitives from which heuristics are generated. In our experiments, these primitives used by Snuba are exactly the same as the ones used by us, e.g. in text experiments we used the same vocabulary to generate unigram based heuristics via the same procedure. Thus, the start up cost to Snuba and IWS is equivalent for all our experiments.
>
> ### Q3b: IWS-LSE top 100
> We have now clarified this in our submission and updated all plots. In short, at each iteration the “top 100” refers to the number of unqueried LFs (LFs not previously shown to user) that could be added to the final set according to Eq. (6), in addition to queried LFs that have $u=1$. Thus, the size of the estimated final set may grow with each iteration,
>
> ( Response continued below -> )

---

> > ### Author Response · Authors · 2020-11-17
> > **Response to AnonReviewer2, continued**
> >
> > ### Q3c Additional Experiments
> > We have carried out additional experiments on a greater range of data types and labeling function families. In particular, we have extended our experiments to image applications:
> > - LFs for images with captions: We have added two experiments on image captions as performed in the Snuba paper. This provides an additional non-sentiment classification task.
> > - LFs for images: We have added two experiments on image data, using LFs that directly operate on the images, without any external modality such as text captions.
> >
> > Please note that these results are preliminary and we are still working on obtaining a full set of baseline comparison for all experiments.

---

> > > ### Author Response · Authors · 2020-11-24
> > > **Additional Update to Submission (User Study on Image Data)**
> > >
> > > As part of our previous update we had added experiments on image data, including the use of image based labeling functions. Now, our second update includes an additional user-study for the new image-based labeling functions on an image classification task. We show results of this user study in Figure 5 in Appendix B.1 (page 13) of our submission. The results are consistent with the findings of our other user experiments on text data included in the original submission. The results show that users are able to distinguish accurate vs. inaccurate labeling functions well, and that the full IWS procedure with real users achieves similar performance as the ones using a simulated oracle.
> > > In addition to these new user experiments, we were also able to evaluate Snuba on image labeling functions using the same kNN image primitives as IWS, and we updated Figure 4 accordingly.

---

### Official Review · AnonReviewer3 · 2020-10-29
**Interesting problem and potentially impactful solution**

**Rating:** 8
**Confidence:** 4

**Review:**

The paper argues that identifying weak supervision signals may require domain expert knowledge and creativity. Therefore, the authors propose an interactive weak supervision solution, in which the annotators assess the quality of automatically identified labelling functions. The proposed solution shows superior performance on a number of classification benchmarks after some minimum number of iterations.

Strengths:
1. The paper studies an important and novel problem in the domain of weak supervision
2. The proposed solution outperforms the baselines
3. The authors show some promising results by running experiments with real human annotators.
4. The paper is well written and easy to follow.

Weaknesses:
1. A large body of work on weak supervision has been completely ignored by the authors. These are some relevant papers:
- Using weak supervision beyond classification (for ranking): https://dl.acm.org/doi/10.1145/3077136.3080832
- Learning from multiple weak supervision signals: https://dl.acm.org/doi/10.1145/3209978.3210041
- Some theoretical justifications for weak supervision training (for ranking): https://dl.acm.org/doi/10.1145/3234944.3234968

2. Fidelity-weighted learning (ICLR 2018) was proposed to automatically identify the quality of weak supervision signals. It is different from the proposed solution in the sense that a small set of labeled instances are used to assess the quality of weak supervision signals as opposed to labeling the weak supervision functions themselves. However, the goal is the same and some comparison (or at least some discussions) are necessary for the paper, I believe.

3. There is no discussion on going beyond classification tasks and even some extreme classification scenarios.

4. I think there should be some difficulty estimation associated with each weak supervision annotation function. Some functions may not be easily assessed by human annotators and this may influence the model. I think this can be addressed in future work.

---

> ### Author Response · Authors · 2020-11-17
> **Response to AnonReviewer3**
>
> Thank you for your thoughtful comments and questions, which we respond to below.
>
> We would like to draw your attention to our updated experiments. We now include a greater range of applications on additional classification tasks, data types, and labeling function types:
> -  Two experiments on a multi-modal dataset of images and image captions previously used in Snuba (Varma et al. 2018) where labeling functions are applied to text captions and a classifier is trained on images.
> -  Two experiments on a dataset for image classification where both the labeling functions and the end classifier are applied on the images directly.
>
> Please note that these results are preliminary and we are still working on obtaining a full set of baseline comparison for all experiments.
>
> ### Q1 related work in information retrieval
> We appreciate the pointers to related literature on weak supervision that we missed. We have added references to all papers to our related work section in our updated draft. Note that, for convenience, most of the changes in our updated paper are denoted by blue text.
>
> ###  Q2 Fidelity Weighted Learning (FWL)
> Thank you for pointing out this interesting work. In our view, FWL offers insight into how one might approach further increasing the quality of noisy labels obtained via our approach, which is an important connection. Our work and Snuba aim to find multiple (generated) weak supervision sources of good quality, in our case without labeled samples. FWL instead refines the quality of weak labels on a per sample basis, and seems to implicitly assume that one or multiple weak supervision sources of sufficient quality have already been selected. We have included a short discussion about the relation of FWL to our work in Section 2 of our paper.
>
> ###  Q3 and Q4
> We have updated our conclusion to acknowledge and comment on these issues.

---

> > ### Author Response · Authors · 2020-11-24
> > **Additional Update to Submission (User Study on Image Data)**
> >
> > As part of our previous update we had added experiments on image data, including the use of image based labeling functions. Now, our second update includes an additional user-study for the new image-based labeling functions on an image classification task. We show results of this user study in Figure 5 in Appendix B.1 (page 13) of our submission. The results are consistent with the findings of our other user experiments on text data included in the original submission. The results show that users are able to distinguish accurate vs. inaccurate labeling functions well, and that the full IWS procedure with real users achieves similar performance as the ones using a simulated oracle.
> > In addition to these new user experiments, we were also able to evaluate Snuba on image labeling functions using the same kNN image primitives as IWS, and we updated Figure 4 accordingly.

---

### Decision · Program_Chairs · 2021-01-07
**Final Decision**

**Decision:**

Accept (Poster)

**Comment:**

The paper proposes a user-interaction framework where users choose a subset of LFs from a family of LFs generated using some template (e.g. keywords for text classification).   The proposed criteria is not very surprising, but the authors present a practical and useful system that is well demonstrated both in the paper and the very careful author feedback.  These enhancements have also been incorporated in the revised version.

 Apart from the literature pointed by the reviewers, here are some more papers that are related to this paper:
1. Gregory Druck, Burr Settles, Andrew McCallum:
Active Learning by Labeling Features. EMNLP 2009: 81-90

2. 	Gregory Druck, Gideon S. Mann, Andrew McCallum:
Learning from labeled features using generalized expectation criteria. SIGIR 2008: 595-602

 3. Data Programming using Continuous and Quality-Guided Labeling Functions. In AAAI, 2020.